# Neoadjuvant chemo-immunotherapy with camrelizumab plus nab-paclitaxel and cisplatin in resectable locally advanced squamous cell carcinoma of the head and neck: a pilot phase II trial

Neoadjuvant chemoimmunotherapy has emerged as a potential treatment option for resectable head and neck squamous cell carcinoma (HNSCC). In this single-arm phase II trial (NCT04826679), patients with resectable locally advanced HNSCC (T2–T4, N0–N3b, M0) received neoadjuvant chemoimmunotherapy with camrelizumab (200 mg), nab-paclitaxel (260 mg/m$^2$), and cisplatin (60 mg/m$^2$) intravenously on day one of each three-week cycle for three cycles. The primary endpoint was the objective response rate (ORR). Secondary endpoints included pathologic complete response (pCR), major pathologic response (MPR), two-year progression-free survival rate, two-year overall survival rate, and toxicities. Here, we report the perioperative outcomes; survival outcomes were not mature at the time of data analysis. Between April 19, 2021 and March 17, 2022, 48 patients were enrolled and received neoadjuvant therapy, 27 of whom proceeded to surgical resection and remaining 21 received non-surgical therapy. The ORR was 89.6% (95% CI: 80.9, 98.2) among 48 patients who completed neoadjuvant therapy. Of the 27 patients who underwent surgery, 17 (63.0%, 95% CI: 44.7, 81.2) achieved a MPR or pCR, with a pCR rate of 55.6% (95% CI: 36.8, 74.3). Treatment-related adverse events of grade 3 or 4 occurred in two patients. This study meets the primary endpoint showing potential efficacy of neoadjuvant camrelizumab plus nab-paclitaxel and cisplatin, with an acceptable safety profile, in patients with resectable locally advanced HNSCC.

Head and neck cancer is the eighth most commonly diagnosed cancer globally, with an estimated 870,000 new cases and 440,000 deaths in 2020[1]. Head and neck squamous cell carcinomas (HNSCC) originate in the mucosal epithelium of the oral cavity, pharynx, and larynx and are the most common cancer types of the head and neck. The head and neck region is anatomically highly complex and serves the primary vital and social functions (such as eating, speaking, and breathing). Multidisciplinary treatments, integrating surgery, chemotherapy, and radiation, aim to maximize treatment effects but have a significant functional impact. Despite that, patients with locally advanced HNSCC carry a high risk of local recurrence (-15–40%) and distant metastasis, with a 5-year overall survival (OS) rate of only 50%[2]. New treatment

✉ e-mail: zxr-850@163.com; lizhm@sysucc.org.cn; liuxk@sysucc.org.cn

options that can improve survival or allow for deintensification of the standard of care are needed.

Neoadjuvant chemotherapy may confer the benefits of organ preservation, local and distant failure reduction, and treatment deintensification in a subgroup of patients with locally advanced HNSCC[3], yet its value has not been fully clarified. In patients with advanced oral cancer, neoadjuvant chemotherapy reduced locoregional recurrence but did not improve survival[4,5]. When neoadjuvant chemotherapy is considered, TPF (docetaxel, cisplatin, and 5-FU) is the preferred regimen[6]. A meta-analysis of the pooled data from five randomized trials confirmed the superiority of induction chemotherapy with TPF over cisplatin plus 5-FU (PF)[7]. However, the most important clinical concern of induction chemotherapy with TPF is the increased overall toxicity, which may compromise the subsequent treatment and efficacy[8]. Crucially, TPF regimens should be administered by experienced physicians who are familiar with the necessary protocols and procedures to ensure safety of patients and maximize adherence throughout the treatment. Besides, dose adaption of TPF may be needed to minimize toxicity, particularly in Asia[3]. This has led to extensive studies of alternative regimens, and most accepted of these are cetuximab plus carboplatin and paclitaxel[9] and cetuximab plus docetaxel and cisplatin[10,11]. With the emergence of new drugs, various combinations have been studied in the neoadjuvant therapy of HNSCC to better preserve the functions of vital organs, reduce systemic toxicity, improve the quality of life, and prolong survival.

Neoadjuvant therapy with immune checkpoint inhibitors has been demonstrated to be promising in various types of cancer, such as melanoma[12,13], non-small cell lung cancer[14], and bladder cancer[15]. Programmed cell death 1 (PD-1) inhibitors nivolumab and pembrolizumab have been approved for the treatment of recurrent/metastatic HNSCC, with a prolonged OS compared with chemotherapy[16–18]. Use of the PD-1 inhibitors in earlier-stage cancers may be more effective given a less evolved tumor and less suppressed immune system[19,20]. Previous preclinical studies supported anti-PD-1 therapy in the neoadjuvant setting rather than in the adjuvant setting[5,21], presumably because of the critical role of bulky tumors emerging during treatment that may contribute to the persistence and activity of antitumor T cells. Neoadjuvant immunotherapies, either alone or in combination with other agents, have been explored in numerous ongoing clinical trials[22–24].

Combinations of chemotherapy with immunotherapy have been demonstrated with favorable clinical outcomes in various tumors[25–27].

In several tumor entities, chemotherapy could induce PD-L1 expression in tumor cells, thereby sensitizing tumor cells to subsequent immunotherapy[28–30]. In addition, chemotherapy could cause immunogenic cell death, which in turn activates antitumor immunity[31,32].

Therefore, we hypothesized that neoadjuvant chemoimmunotherapy may be a promising therapeutic option for potentially resectable locally advanced HNSCC. This phase II study evaluated the efficacy and safety of neoadjuvant chemoimmunotherapy with camrelizumab plus nab-paclitaxel and cisplatin (NeoCPC) in patients with locally advanced HNSCC. Besides, potential biomarkers for predicting response to NeoCPC were also explored, including *HPV* status, PD-L1 expression, genomic profiling, tumor mutational burden (TMB), and tumor infiltration immune cells.

In this work, NeoCPC shows potential efficacy in terms of radiographic and pathologic response, with an acceptable safety profile, in patients with resectable locally advanced HNSCC. Long-term follow-up is still ongoing.

## Results

### Patient characteristics

Between April 19, 2021, and March 17, 2022, 48 patients were enrolled (Fig. 1). Demographics and clinical characteristics are summarized in Table 1. The median age was 59 (range: 27–73) years, and 42 (87.5%) patients were male. One patient aged 73 years old who was in good general condition and met the eligible criteria was also included. Thirty-three (68.8) patients had a smoking history, and 24 (50%) had an alcohol consumption history. Of the 48 patients, 16 (33.3%) had oral cancer, 14 (29.2%) had oropharyngeal cancer, 9 (18.8%) had laryngeal cancer, and 9 (18.8%) had hypopharyngeal cancer. Nine patients were HPV-positive (18.8%), seven had oropharyngeal carcinoma, and two had laryngeal carcinoma. Most patients had stage IV disease (79.2%) and lymph node involvement (91.7%).

### Treatment characteristics

All 48 patients completed three cycles of NeoCPC. Twenty-seven (56.2%) patients underwent surgical resection, including 15 oral cancer (15 radical surgery), nine oropharyngeal cancer (seven radical surgery and neck lymph node dissection and two simple tonsil resection), two hypopharyngeal cancer (one total laryngectomy and lymph node dissection and one simple neck lymph node dissection) and one laryngeal cancer (simple neck lymph node dissection). One patient with oral cancer (Patient 19) refused surgery for personal and economic reasons

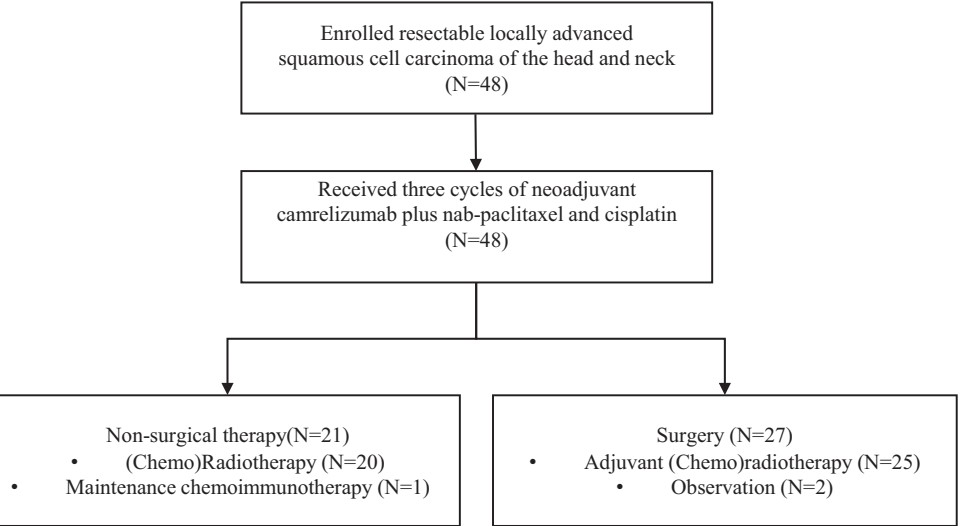

**Fig. 1 | Study flow chart.** In total, 48 patients were enrolled in this trial and received neoadjuvant therapy; among them, 21 patients received adjuvant therapy without surgery, 27 patients underwent surgical resection.

**Table 1 | Demographic and clinical characteristics of the patients**

| Characteristics | Patients (n = 48) |
|---|---|
| Age, median (range), years | 59 (27, 73) |
| **Sex, n (%)** | |
| Male | 42 (87.5) |
| Female | 6 (12.5) |
| **Smoking, n (%)** | |
| No | 15 (31.2) |
| Yes | 33 (68.8) |
| **Alcohol use, n (%)** | |
| Never | 24 (50.0) |
| Ever | 24 (50.0) |
| **Betel nut, n (%)** | |
| No | 45 (93.8) |
| Yes | 3 (6.3) |
| **Subtypes, n (%)** | |
| Oral cancer | 16 (33.3) |
| Oropharyngeal cancer | 14 (29.2) |
| Laryngeal cancer | 9 (18.8) |
| Hypopharyngeal cancer | 9 (18.8) |
| **HPV status, n (%)** | |
| Positive | 9 (18.8) |
| Negative | 39 (81.2) |
| **Clinical T stage, n (%)** | |
| T2 | 9 (18.8) |
| T3 | 21 (43.8) |
| T4 | 18 (37.5) |
| **Clinical N stage, n (%)** | |
| N0 | 4 (8.3) |
| N1 | 1 (2.1) |
| N2 | 40 (83.3) |
| N3 | 3 (6.3) |
| **Clinical TNM stage, n (%)** | |
| II | 6 (12.5) |
| III | 4 (8.3) |
| IV | 38 (79.2) |
| **PD-L1 combined positive score, n (%)** | |
| <1 | 9 (18.8) |
| 1–19 | 33 (68.8) |
| ≥20 | 5 (10.4) |
| Not evaluated | 1 (2.1) |

after completing NeoCPC and returned to the hospital for surgery 191 days later. The patient's genetic data were not included in the subsequent analyses. Of the 27 resected patients, 25 received postoperative adjuvant radiotherapy (n = 7) or chemoradiotherapy (n = 18), and the remaining two patients refused further treatment.

Meanwhile, 21 patients underwent non-surgical therapy (one oral cancer, five oropharyngeal cancer, eight laryngeal cancer, and seven hypopharyngeal cancer). Among the 21 patients, 3 received definitive radiotherapy (one oropharyngeal cancer, one laryngeal cancer, and one hypopharyngeal cancer), 17 received concurrent chemoradiotherapy (one oral cancer, three oropharyngeal cancer, seven laryngeal cancer, and six hypopharyngeal cancer), and the remaining one patient refused radiotherapy and received maintenance chemoimmunotherapy with camrelizumab plus nab-paclitaxel and cisplatin.

## Efficacy outcomes

In the first stage, 9 out of 11 (81.8%) initial patients showed an objective response, exceeding the threshold required, and thus the study continued to the second stage. Among the 48 patients finally enrolled, 10 achieved a CR (one oral cancer, four oropharyngeal cancer, two laryngeal cancer, and three hypopharyngeal cancer), and 33 achieved a PR (13 oral cancer, eight oropharyngeal cancers, seven laryngeal cancer, and five hypopharyngeal cancer), with an ORR of 89.6% (95% CI: 80.9, 98.2] (Fig. 2). Four (8.3%) patients had an SD (one oral cancer, two oropharyngeal cancer, and one hypopharyngeal cancer). One (2.1%) patient with oral cancer had a PD.

Of the 27 resected patients, 17 (63.0%, 95% CI: 44.7, 81.2) achieved an MPR or pCR, with a pCR rate of 55.6% (95% CI: 36.8, 74.3) (including seven oral cancer, six *HPV*-positive oropharyngeal cancer, one *HPV*-negative oropharyngeal cancer, and one laryngeal cancer). Twenty-one out of the 27 patients had a clinical to pathological downstaging (77.8%, 95% CI: 62.1, 93.5). Supplementary Fig. 1A, B shows a representative example of a CR in the primary tumor and metastatic lymph nodes in a patient (P37) with T3N2M0 *HPV*-positive oropharyngeal cancer after NeoCPC. The patient was subsequently treated with adjuvant radiotherapy.

Regardless of *HPV* status, pathologic features observed in patients with a significant tumor response included mixed inflammatory infiltration, fibrosis, visible tumor regression, giant cell reaction, calcification, and acellular keratin. Supplementary Fig. 1C, D shows a representative example of a striking response in a patient (P35) with T4aN0M0 oral squamous cell carcinoma. Imaging assessment showed a PR after three cycles of NeoCPC, while the subsequent pathological analysis demonstrated a pCR.

As of the data cutoff (August 30, 2023), the median follow-up time was 666 (IQR: 537, 753) days. Survival outcomes were not mature (Supplementary Fig. 2). The estimated 1-year OS and PFS rates were both 97.9% (95% CI: 86.1, 99.7). Follow-up was still ongoing.

## Safety

Treatment-related adverse events (TRAEs) are summarized in Table 2. The most common TRAEs of any grade were alopecia (100.0%), nausea/vomiting (60.4%), and fatigue (50.0%). TRAEs of grade 3 or worse occurred in two (6.3%) patients. One patient experienced grade 3 peripheral neuropathy with lower extremity numbness, mobility problems, and difficulty walking. After neurotrophic therapy, the patient gradually recovered within three months after NeoCPC. One patient developed grade 3 pneumonitis after the third cycle of NeoCPC, and the symptoms were resolved within two weeks with active antibiotic therapy and supportive care. No previously unknown or unexpected TRAEs were observed. No TRAEs leading to discontinuation of all study drugs, dose reduction, or death occurred. There were no severe immune-related adverse events (irAEs).

No treatment-related surgical delay was observed. The mean interval from the last dose of NeoCPC to surgery was 23.9 (range: 8, 41) days among the 26 patients who underwent planned surgery. The median duration of postoperative hospital stay was 5.6 days, with a minimum of 2 days and a maximum of 16 days. Thirteen of 27 (48.1%) patients experienced at least one postsurgical complication (Supplementary Table 1). The most common complication was swelling in five (18.5%) patients. All patients recovered in about one month after symptomatic treatment. Postoperative hemorrhage occurred in three patients, and appropriate treatment was administered, including local pressure and hemostatic drugs. The swelling gradually decreased, with no obvious increase in bleeding. Delayed wound healing occurred in two patients with oral cancer after flap repair, and the symptoms resolved ~1 month after anti-infection therapy and enhanced nutritional support.

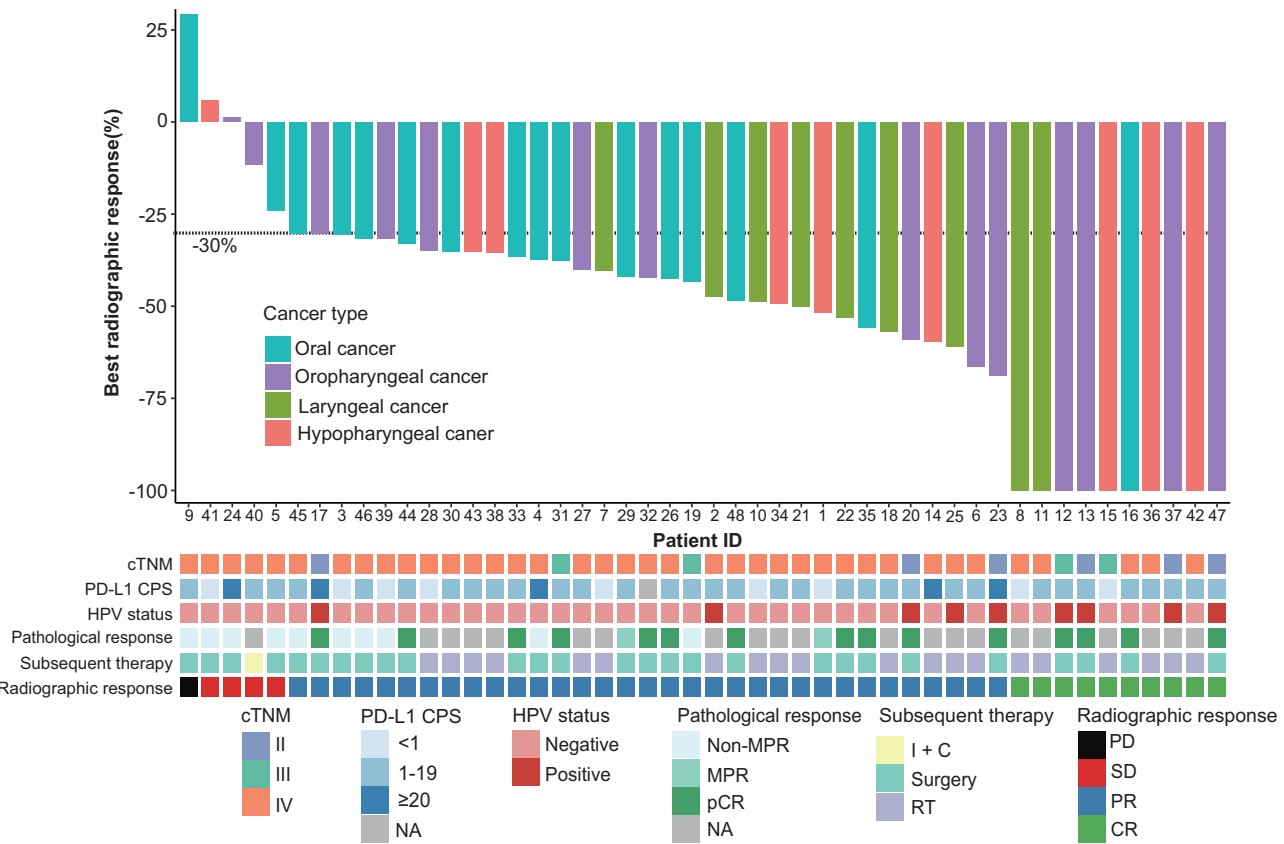

**Fig. 2 | The waterfall plot of best radiographic response by RECIST 1.1 ($n = 48$).** Each bar indicates one patient. Source data are provided as a Source Data file.

## Consistency between radiographic and pathologic response

Among 27 patients who underwent surgery, 26 were evaluated for the consistency between radiographic and pathologic response. According to RESIST v1.1, all patients who achieved a CR got a pCR, whereas no patients with an SD got a pCR (Fig. 3a). Patients with an ORR were more likely to get a pCR ($P = 0.022$, Fig. 3b).

### Table 2 | Treatment-related adverse events (TRAEs)

| TRAEs, n (%) | Any grade | Grade 1–2 | Grade 3–4 |
|---|---|---|---|
| Alopecia | 48 (100.0) | 48 (100.0) | 0 |
| Nausea/vomiting | 29 (60.4) | 29 (60.4) | 0 |
| Fatigue | 24 (50.0) | 24 (50.0) | 0 |
| Peripheral neuropathy | 16 (33.3) | 15 (31.3) | 1 (2.1) |
| Pain (lymph node, oral)[a] | 15 (31.3) | 15 (31.3) | 0 |
| RCCEP | 14 (29.2) | 14 (29.2) | 0 |
| Skin (rash, dryness, dermatitis)[b] | 10 (20.8) | 10 (20.8) | 0 |
| Anemia | 7 (14.6) | 7 (14.6) | 0 |
| Constipation | 6 (12.5) | 6 (12.5) | 0 |
| Fever | 5 (10.4) | 5 (10.4) | 0 |
| Leukopenia | 5 (10.4) | 5 (10.4) | 0 |
| Thrombocytopenia | 4 (8.3) | 4 (8.3) | 0 |
| Oral mucositis/lichenoid reaction/dry mouth/dysphagia | 3 (6.3) | 3 (6.3) | 0 |
| Pneumonitis | 2 (4.2) | 1 (2.1) | 1 (2.1) |
| Diarrhea | 2 (4.2) | 2 (4.2) | 0 |

*AE* adverse event, *RCCEP* reactive cutaneous capillary endothelial proliferation.
[a]Including myalgia and arthralgia.
[b]Including papulopustular rash and all skin and subcutaneous tissue disorders.

## Post hoc biomarker analysis and evolution of tumor microenvironment

To explore the potential molecular biomarkers for the radiographic and pathologic response, we performed immunohistochemistry to detect PD-L1 expression, NGS to detect mutations of 437 cancer-related genes, and multiplex immunofluorescence to detect immune cell infiltration in pretreatment tumor biopsies from 47 patients and post-treatment surgical samples from 25 patients.

### Potential biomarkers for radiographic response

The PD-L1 test was performed in all patients except one whose specimens were not qualified. Thirty-eight of 47 (97.9%) patients were PD-L1 positive (CPS ≥ 1), of which 33 (86.8%) had a CPS of 1–19 and five (13.2%) had a CPS of ≥20. For genetic analysis, the most frequently mutated genes were *TP53* (77.1%), *CDKN2A* (33.3%), *FAT1* (25%), *CCND1* (22.9%), and *NOTCH1* (22.9%) (Fig. 4a). The median TMB was 3.15 mutations/MB, similar to a previous report[33]. The distribution of PD-L1 expression was significantly associated with the different sites of tumor origins ($P = 0.036$). There was no significant difference in the distributions of TMB or PD-L1 expression levels according to TNM staging or in the distribution of PD-L1 expression according to anatomic site (Supplementary Fig. 3). No significant difference was observed in the PD-L1 expression, *HPV* status, somatic variants of 437 genes, and TMB between ORR (CR + PR) and non-ORR (SD + PD) groups. Similarly, no significant difference was noted between CR and non-CR (PR + SD + PD) groups.

Significantly, a higher radiographical response rate was observed in *HPV*-positive patients ($P = 0.012$, Fig. 4b). Meanwhile, a lower radiographical response rate was observed in patients with altered *TP53* ($P = 0.006$, Fig. 4c) and those with altered *TERT* ($P = 0.01$, Fig. 4d). The correlation analyses of *TP53*/*TERT* alternations with *HPV* were further performed. As shown in Fig. 4e, the *TP53* alternation was more

a

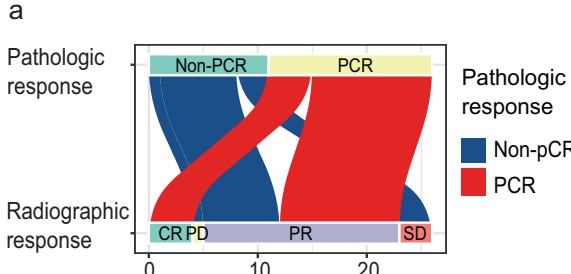

b

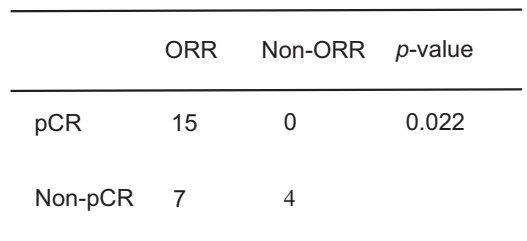

| | ORR | Non-ORR | *p*-value |
|---|---|---|---|
| pCR | 15 | 0 | 0.022 |
| Non-pCR | 7 | 4 | |

**Fig. 3 | Correlation analysis of radiographic response and pathologic response (*n* = 26).** Sankey plot shows the relationship between radiographic response and pathologic response (**a**). The consistency between radiographic response and pathologic response analyzed by using the two-sided Fisher exact test (**b**). Source data are provided as a Source Data file.

common in *HPV-* patients (*P* < 0.001), while the *TERT* alternation was also more common in *HPV-* patients, although the difference did not reach statistical significance (*P* = 0.16). The correlations of radiographical tumor response with *TP53/TERT* alternations were further carried out among the *HPV−* patients. *TP53* alternation was not significantly associated with radiographic tumor response (*P* = 0.807, Fig. 4f). A lower radiographic response rate was observed in patients with altered *TERT* (*P* = 0.049, Fig. 4g).

The density of CD8 + T cells in the tumor area was significantly associated with the percentage of radiographic response (*P* = 0.0005, Fig. 5a), and so were the densities of M1-like macrophage cells (*P* = 0.009, Fig. 5b) and CD4 + T cells (*P* = 0.021, Supplementary Fig. 4A), respectively. The representative images of multiplex immunofluorescence are shown in Fig. 5c, d and Supplementary Fig. 4B. However, no significant association was found in the other tumor-infiltering immune cell subsets, including M2-like macrophage cells, CD56bright NK cells, CD56dim NK cells, and T regulatory (Treg) cells (Supplementary Fig. 5). Patients with *HPV* infection harbored higher levels of CD8 + T cells and M1-like macrophages (Fig. 5e). In comparison, those with no *TP53* or *TERT* alternation harbored greater levels of CD8 + T cells and M1-like macrophages (Fig. 5f, g). Further exploratory analysis showed no significant correlations between CD8 + T cells or M1-like macrophages and radiographical tumor response in *HPV*-positive and *HPV*-negative patients.

### Potential biomarkers for pathologic response
No significant genetic differences were noted between pCR and non-pCR patients by using the Fisher exact test. The PD-L1-positive expression and *HPV* infection were significantly enriched in patients with pCR. All pCR patients were PD-L1-positive (Fig. 6a), and all non-pCR patients were *HPV*-negative (Fig. 6b). The *TP53*-altered patients were less likely to achieve a pCR (Fig. 6c).

A higher median CD8+ intratumoral T-cell density was observed in patients with pCR compared to those without pCR, although this difference was not statistically significant. There was no significant difference in the densities of M1-like macrophage cells, M2-like macrophage cells, CD56bright NK cells, and CD56dim NK cells in the tumor area between the pCR and non-pCR patients. However, the CD8 + T- cell density in the tumor stroma was much higher (Fig. 6d).

### Change in immune cell subsets following NeoCPC
M1-like macrophage cells in the tumor stroma were significantly increased after treatment in patients with pCR (*P* = 0.0234), but not in those with non-pCR (*P* > 0.05). The other types of tumor infiltration immune cells were not significantly changed (Supplementary Fig. 6).

## Discussion
In this study, neoadjuvant camrelizumab combined with nab-paclitaxel and cisplatin was safe and feasible for resectable locally advanced HNSCC. The ORR was 89.6%, and the toxicities were generally manageable. The Grade 3 or 4 TRAEs occurred in 6.3% of the patients, lower than that previously reported (21.7%)[34] (9.9%)[23]. Most of the TRAEs were chemotherapy-related. The dose of cisplatin in our treatment regimen is relatively lower (60 mg/m²) than that of the previous study (100 mg/m²)[35]. Overall, our protocol is mild, with few side effects and high tolerance among patients. A delay in surgery following neoadjuvant chemotherapy may affect oncologic outcomes in HNSCC[36]. Neoadjuvant chemotherapy did not appear to increase perioperative morbidity among patients undergoing surgery for HNSCC[37]. We did not observe a treatment-related surgery delay or any concerning effects on surgical outcomes. Larger studies are needed to determine whether neoadjuvant chemotherapy in combination with immunotherapy may compromise surgery.

The NeoCPC in locally advanced HNSCC showed encouraging efficacy outcomes, including a high ORR rate of 89.6%. Multiple studies have reported that immunotherapy combined with chemotherapy can obtain a higher ORR rate in HNSCC[23,24,34]. Zhang et al. reported an ORR of 96.7% (29/30) in patients with locally advanced HNSCC receiving neoadjuvant camrelizumab plus chemotherapy[23]. This may be because of a comparatively higher cisplatin dose of 75 mg/m² (60 mg/m² in this study). Also, their study almost exclusively included patients with tumors in the throat (oropharynx + larynx + hypopharynx), which are more sensitive to chemotherapy. In this study, the ORR was 87.5% for oral cancer and 90.6% for throat cancers. In addition, the proportion of patients with stage IV disease (73.3%) was lower than that in our group (79.2%). Surgical resection is often preferred for locally advanced HNSCC[2]. A high ORR rate after neoadjuvant therapy means a lower tumor burden that is suitable for surgery. The scope of surgical resection is to be determined according to the image before neoadjuvant therapy. Whether the image after treatment can be used to adjust the scope of surgery and whether CR patients can have direct radiotherapy without surgery are worthy of further study.

Nab-paclitaxel was used in this study as the part of the chemotherapy regimen, instead of paclitaxel. Induction chemotherapy with a nab-paclitaxel-based regimen has been reported to be associated with better survival outcomes when compared with docetaxel-based regimen[38]. The potential mechanism may involve secreted protein acidic and rich in cysteine (SPARC) that plays a role in albumin receptor-mediated endometrial transport[39] and activated RAS and/or PI3K pathways that are common drivers for initiating micropinocytosis, the process by which macromolecules such as albumin are taken up into cells. SPARC expression was found in tumor and stromal cells but not in the adjacent normal oral mucosa in HNSCC[40] and correlated with better tumor response to nab-paclitaxel in HNSCC patients[41]. Meanwhile, the frequently activated RAS and/or components of the PI3K pathways may also help to explain the enhanced antitumor effect of nab-paclitaxel in HNSCC[42]. Collectively, these data suggest that nab-paclitaxel may be a potential option in induction

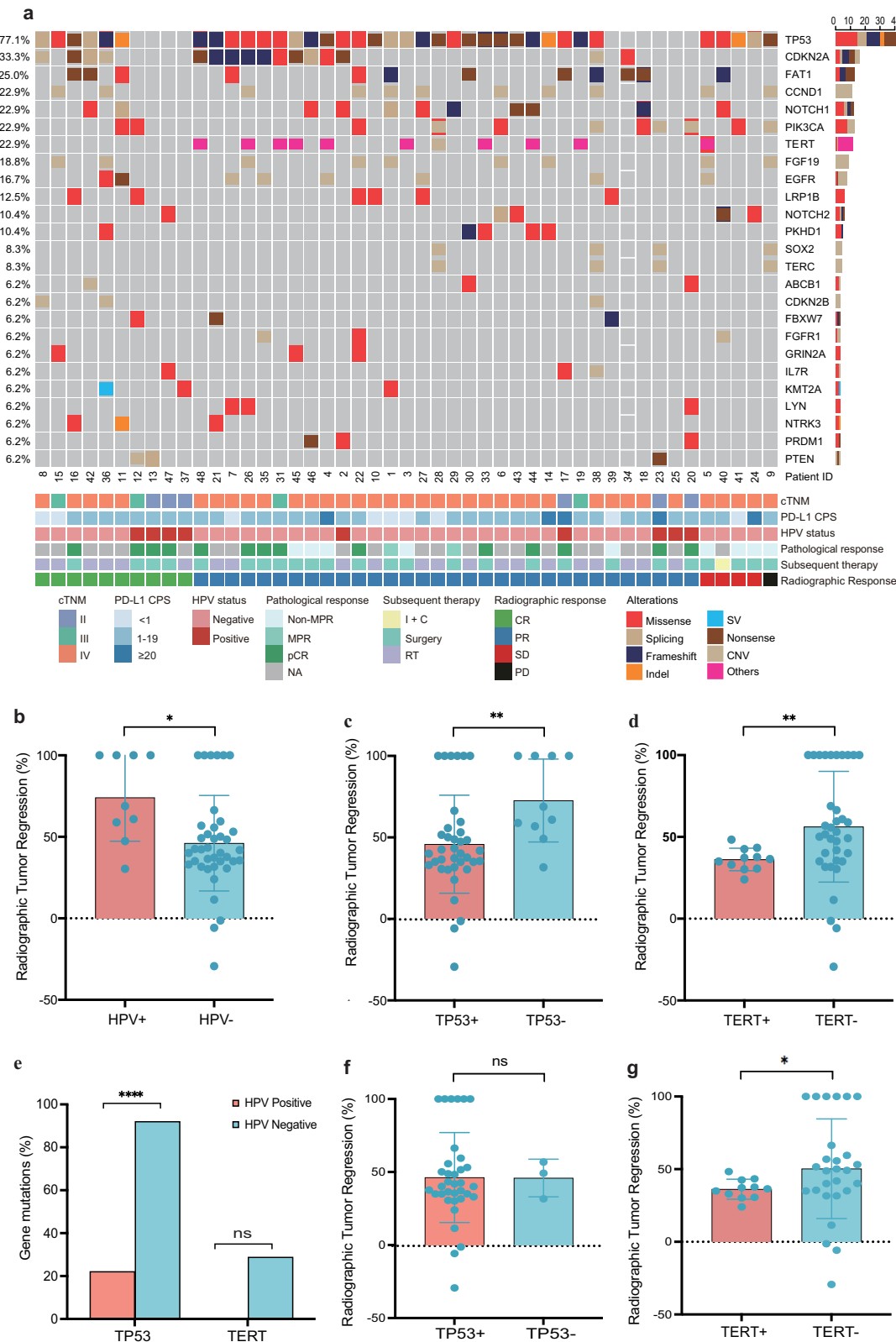

chemotherapy of HNSCC. Future studies are needed to clarify it further.

Three cycles of NeoCPC induced a major or complete pathological response rate of 63.0% (17/27) in surgical patients, with a pCR rate of 55.6% (15/27), which were higher than previous studies of neoadjuvant immune checkpoint blockade in locally advanced HNSCC. The MPR or pCR rate was 5–7% for pembrolizumab alone[43,44], 5.9–17% for nivolumab alone, and 20–35% for nivolumab combined with ipilimumab[22,45,46]. Zhang et al. reported an MPR of 74.1% and a pCR rate of 37% in patients with locally advanced HNSCC receiving neoadjuvant chemoimmunotherapy[23]. The comparatively higher MPR rate observed in their study may be partly because that 21 patients (6 for CR, 14 for PR, and 1 for SD) in our group did not receive surgical treatment, so we could not evaluate the pathologic response of these

**Fig. 4 | The genetic landscape and radiographic tumor response analysis.**
Alterations as assessed by next-generation sequencing of baseline primary tumor samples ($n = 47$) (**a**). Comparisons of radiographic tumor response between *HPV*-positive ($n = 9$) and *HPV*-negative ($n = 39$) patients ($P = 0.012$) (**b**); *TP53*-mutant ($n = 37$) and *TP53*-wild-type ($n = 10$) patients ($P = 0.006$) (**c**); and *TERT*-mutant ($n = 11$) and *TERT*-wild-type ($n = 36$) patients by using the two-sided Mann–Whitney *U* nonparametric test ($P = 0.01$) (**d**). Correlation between *HPV* status (HPV-positive: $n = 9$; HPV-negative: $n = 39$) and *TP53* ($n = 37$, $P < 0.0001$) or *TERT* mutations ($n = 11$,

$P = 0.16$) by using the two-sided Mann–Whitney *U* nonparametric test (**e**). Comparisons of radiographic tumor response between *TP53*-mutant ($n = 35$) and *TP53*-wild-type ($n = 3$) patients ($P = 0.807$) (**f**); and *TERT*-mutant ($n = 11$) and *TERT*-wild-type ($n = 27$) patients ($P = 0.049$) (**g**) in *HPV*-negative patients by using the two-sided Mann–Whitney *U* nonparametric test. The histogram plots show the mean values of radiographic tumor response and standard deviation (SD). The dot represents an individual data point. *$P < 0.05$, **$P < 0.01$, ***$P < 0.001$, ****$P < 0.0001$, ns no significance. Source data are provided as a Source Data file.

patients after NeoCPC treatment. Our results showed that the MPR was correlated with the ORR in image evaluation. Therefore, the MPR rate we studied should be higher. Huang et al. reported pathologic response rates of 16.7% (pCR) and 27.8% (MPR) with gemcitabine and cisplatin combined with toripalimab. Patients in their study received only two cycles of neoadjuvant therapy followed by surgery.

The correlations between pathological remission after neoadjuvant chemoimmunotherapy and radiographical response, as well as their prognostic values have not been conclusively concluded. The findings of our study showed a significant correlation between radiographical response and pathological response, consistent with the previous results of neoadjuvant nivolumab[47]. However, the conflicting results of no significant correlation were also reported in patients with resectable HNSCC receiving neoadjuvant chemoimmunotherapy[34,48]. As for the prognostic prediction of pathological response, Trisha et al. added neoadjuvant and adjuvant pembrolizumab to (chemo)radiotherapy in patients with previously untreated, resectable local-regionally advanced HNSCC. Patients with a (partial, major, or complete) pathologic response had significantly increased 1-year PFS (93% vs. 72%) and OS (100% vs. 93%) rates compared with those without[44]. Both the 1-year PFS and OS rates of our study were 97.9%. The only patient who died in our study was evaluated as non-MPR after surgery. Further follow-up is ongoing to determine whether pCR or MPR is associated with PFS and OS in our study. In addition, future trials will explore the potential to minimize the extent of surgical resection and the intensity of adjuvant therapy in patients with a probable MPR.

The Groupe Oncologie Radiothérapie Tête et Cou (GORTEC) 2000-01 trail showed a larynx preservation rate of 80% with induction TPF[49]. In the RTOG91-11 study, the average larynx preservation rate of the three different experimental groups was 71.5%[50]. Zhang et al. reported the throat and hypopharyngeal function retention rate was 85.7%[23]. Our results showed a throat and hypopharyngeal function retention rate of 83.3% (15/18), slightly higher than the two studies that used chemotherapy alone. More importantly, NeoCPC showed fewer side effects and a high patient acceptance. Combining immunotherapy with neoadjuvant chemotherapy is a promising means for improving laryngeal function retention, and of course, future trials with larger sample sizes are needed.

In this study, PD-L1 expression and TMB were not significantly correlated with radiographic tumor response, while PD-L1 expression was significantly correlated with pCR. Several studies have been focusing on the identification of biomarkers. Vos et al. reported no statistically significant difference in PD-L1 expression, TMB, and density of CD3 + CD8 + T cells between patients with and without MPR receiving nivolumab or nivolumab plus a single dose of ipilimumab prior to surgery[45]. Ferrarotto et al. revealed that neither baseline CD8 + TIL density nor PD-L1 expression was correlated with overall response in oropharyngeal squamous cell carcinoma patients, but a trend toward greater CD8 + TIL change in MPR patients[51]. The utility of PD-L1 expression to stratify the benefit of neoadjuvant chemoimmunotherapy in HNSCC patients should be further assessed. Besides, biomarker exploration shows high correlations between intratumor CD8 + T-cell and radiographical response. The correlation between CD8 + T cells and the response to immunotherapy has been revealed in

studies across many cancer types, including local advanced and recurrent or metastatic HNSCC[16,51–54].

*HPV* infection, one of the carcinogenic agents of HNSCC, has been identified as a favorable prognostic factor for survival in HNSCC patients treated with standard chemotherapy and radiotherapy. *HPV*-positive tumors exhibited higher immunogenicity with greater activated CD8 + T-cell infiltration than their *HPV*-negative counterparts by transcriptomic analysis of 280 HNSCCs using the TCGA Database[55]. It may be seen that *HPV* infection could activate immunoreaction. However, it is still largely unknown whether patients with *HPV* infection can benefit more from immunotherapy. In the Keynote 012 study of pembrolizumab in patients with recurrent or metastatic HNSCC, the ORR was 24% in *HPV*-positive patients and 16% in *HPV*-negative patients[17]. In the subgroup analysis of Checkmate 141, patients with p16-positive disease tended to benefit more from nivolumab when compared with the standard therapy (HR: 0.56, 95% CI: 0.32, 0.99)[56]. However, some other studies (KEYNOTE-055 trial, KEYNOTE-040 trial, KEYNOTE-048 trial) showed a similar ORR rate regardless of *HPV* status, and medium PFS and OS were also not different based on *HPV* status[18,52,57,58]. The SITC subcommittee does not recommend using *HPV* status to guide the use of immunotherapy[59,60]. In this study, patients with *HPV*-positive tumors showed more intratumor CD8 + T cells and M1-like macrophage cells and responded much better to neoadjuvant chemoimmunotherapy. These results suggest that patients with *HPV* infection have the potential to benefit from neoadjuvant chemoimmunotherapy. The increased M1-like macrophage cells in the tumor stroma after neoadjuvant chemoimmunotherapy indicate that this regimen may recruit more M1-like macrophage cells to regulate the immune microenvironment for promoting apoptosis of tumor cells.

Many large-scale studies reported the frequently alternated genes in HNSCCs, such as *TP53*, *CDKN2A*, *FAT1*, *NOTCH1*, *PIK3CA*, and *FBXW*. We also detected the alterations in these genes in this study, but did not find any difference between patients with ORR (CR + PR) and non-ORR (SD + PD), CR and non-CR (PR + SD + PD), as well as those with pCR and non-pCR. It is probably limited by sample size and follow-up time. Nevertheless, our data did show a higher percentage of radiographic tumor regression in patients with wild-type *TP53*. Jiang et al. reported that *TP53* alterations predicted worse OS in HNSCC patients receiving immunotherapy[61]. *TERT* gene has been reported to associate with poor PFS and OS in HNSCC, but no direct evidence explains the relationship with immunotherapy.

Interestingly, *TP53* and *TERT* alternations were found to be enriched in *HPV*-negative patients, while more intratumor CD8 + T cells and M1-like macrophage cells were found in *HPV*-positive patients. That is, the same *HPV*-positive population is also *TP53/TERT* wildtype, while has high levels of CD8 and M1-like macrophage tumor microenvironment. To better understand the potential role of *HPV* in response correlation of the biomarkers, further correlation analysis of radiographical response with *TP53/TERT* alternation was performed in the *HPV*-negative and/or *HPV*-positive populations. There were no significant differences in radiographic response regarding *TP53/TERT* alternation in the *HPV*-negative patients, or intratumor CD8 + T cells and M1-like macrophage cells in *HPV*-positive and *HPV*-negative patients, suggesting that their radiographical response correlations

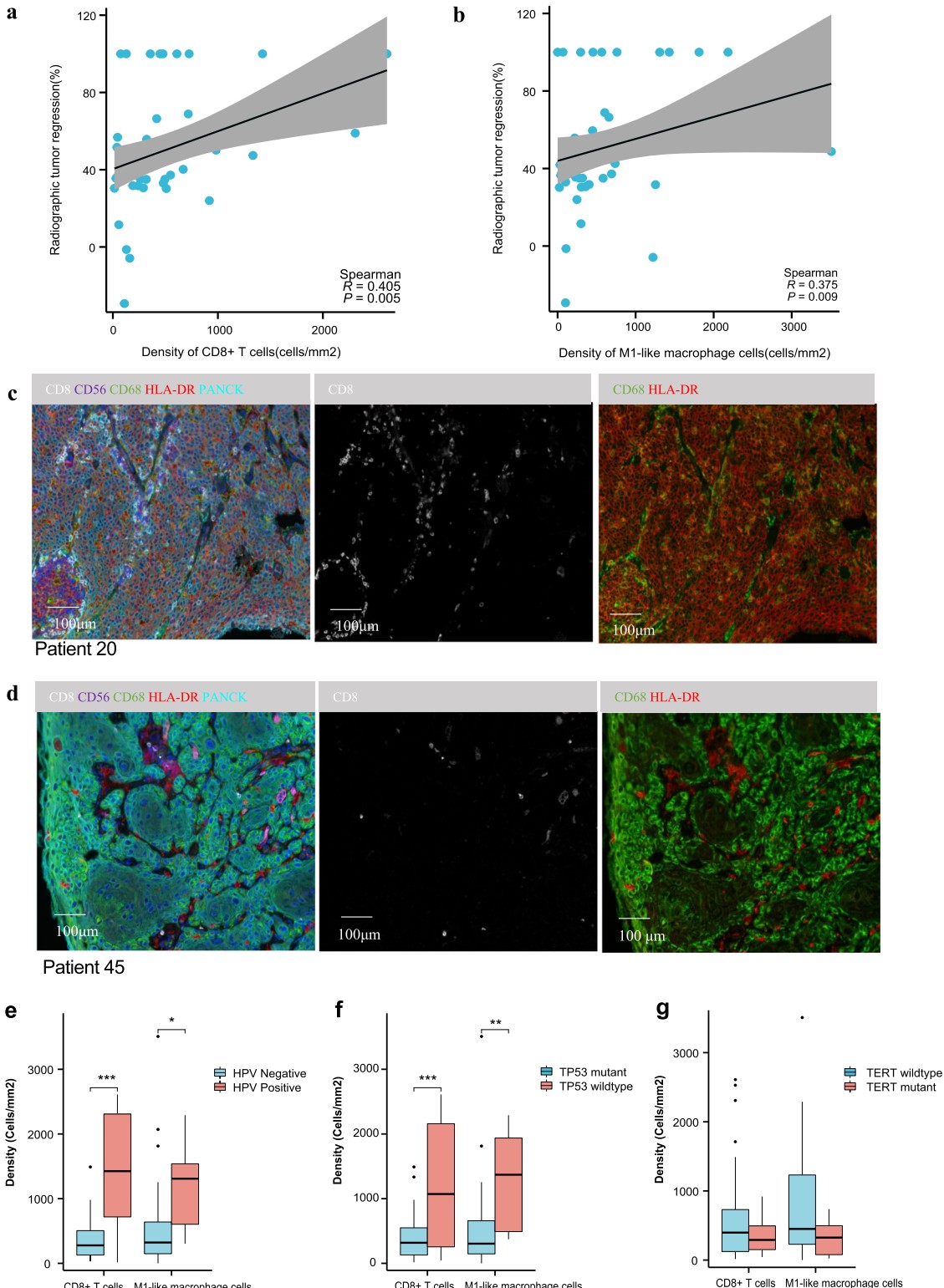

**Fig. 5 | Tumor-infiltrating lymphocyte analysis.** The radiographic tumor response was significantly associated with the densities of tumor-infiltrating CD8 + T cells ($n$ = 47, Spearman $R$ = 0.405, $P$ = 0.0005) (**a**) and tumor-infiltrating M1-like macrophage cells ($n$ = 47, Spearman $R$ = 0.375, $P$ = 0.009) (**b**) by using the two-sided Spearman correlation text. The line represents a fitted line. The gray shadow represents the corresponding 95% confidence interval. The baby-blue dot represents an individual data point. The multiplex immunofluorescence images of the tumor sites in patient 20 (**c**) and patient 45 (**d**). Primary antibodies targeting CD8, CD56, CD68, HLA-DR, and Pan-CK were used on the same slide. Comparisons of the densities of tumor-infiltrating CD8+ cells ($P$ < 0.0001) and M1-like macrophage cells

($P$ = 0.01) between *HPV*-positive ($n$ = 9) and *HPV*-negative ($n$ = 38) patients (**e**); CD8+ cells ($P$ = 0.0002) and M1-like macrophage cells ($P$ = 0.0028) between *TP53*-mutant ($n$ = 37) and *TP53*-wild-type ($n$ = 10) patients (**f**), and CD8+ cells ($P$ = 0.19) and M1-like macrophage cells ($P$ = 0.07) between *TERT*-mutant ($n$ = 11) and *TERT*-wild-type ($n$ = 36) patients (**g**) analyzed by using the two-sided unpaired $t$ test. Box-and-whisker plots show the distribution (box, whiskers, and outliers). The center line represents the median, the box limit represents the interquartile range (IQR), the whiskers represent the 1.5×IQR, and the outliers represent an individual data point. *$P$ < 0.05, **$P$ < 0.01, ***$P$ < 0.001, ns no significance. Source data are provided as a Source Data file.

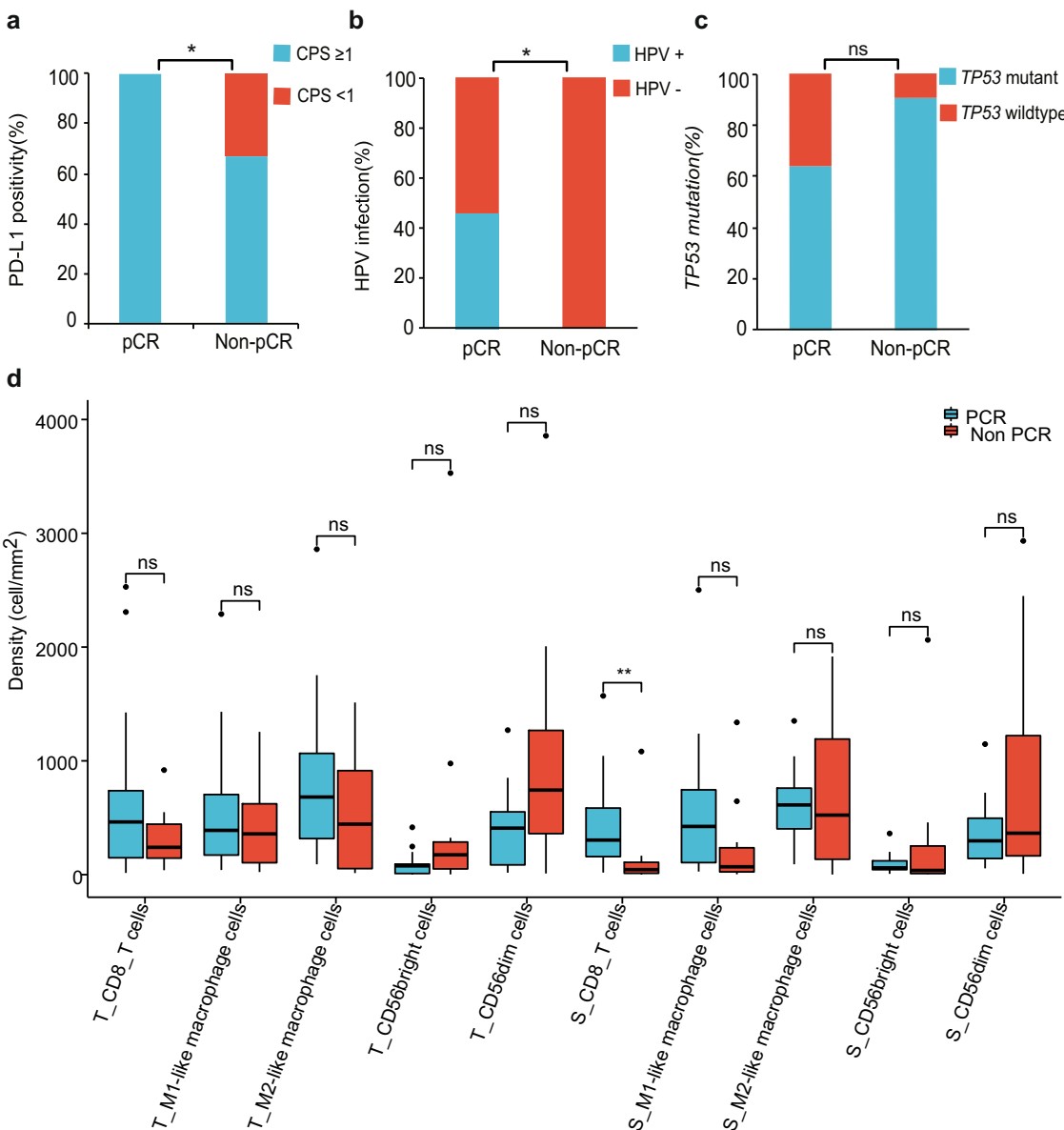

**Fig. 6 | Exploratory analysis of pathologic response characteristics.** The percentages of patients with *HPV* infection (**a**, *P* = 0.03), PD-L1 positive expression (**b**, *P* = 0.046), or *TP53* mutation (**c**, *P* = 0.18) in the pCR (*n* = 15) and non-pCR (*n* = 11) patients were compared by using the two-sided Fisher exact test. Comparison of the density of tumor-infiltering immune cells in patients with pCR (*n* = 15) and non-pCR (*n* = 11) by using the two-sided Wilcoxon rank-sum test (**d**). Box-and-whisker plots show the distribution of cell density. The center line represents the median, the box limit represents the interquartile range (IQR), the whiskers represent the 1.5×IQR, and the outliers represent individual data point. *\*P* < 0.05, ns no significance. Source data are provided as a Source Data file.

may be biased to some extent by *HPV* mutation of patients. Unfortunately, the sample size is small in this study, for example, there were only nine *HPV*-positive patients and three *HPV*-negative patients being *TP53* wildtype, which precluded the possibility for us to explore the potential correlation of *TP53/TERT* alternation, immune infiltrating cells, and *HPV* status in HNSCC. Whether the correlations of these biomarkers with radiographic response was biased by *HPV* status of patients still needs further investigation.

Certain limitations must be acknowledged. First, this is a single-center, single-arm study with a small sample size, and the statistical power is limited. In addition, the follow-up time is not long enough, and survival outcomes are still pending. Assessment of MPR in the neoadjuvant setting of HNSCC is not well defined and was based on the criteria of lung cancer or breast cancer. The criteria specific to HNSCC should be developed in the future. Also, exploratory analysis was done in patients with ORR versus those who had non-ORR, yet only five

patients were included in the non-ORR group (four SDs and one PD), which may make the comparisons a bit weak, future studies are needed to determine the prognostic values of PD-L1 CPS and other biomarkers. In addition, multiplex immunofluorescence was limited to only a few cell populations (CD8, M1-like macrophage, CD4, and Treg) due to insufficient tissue samples, particularly for patients with hypopharyngeal or laryngeal cancer. Future studies need to evaluate deeper profiling of exhaustion markers and their pre-and-post changes, so as to better understand the potential tumor microenvironment effects after neoadjuvant chemoimmunotherapy.

In conclusion, neoadjuvant chemoimmunotherapy with camrelizumab plus nab-paclitaxel and cisplatin showed potential efficacy in terms of radiographic and pathological tumor response and larynx preservation, with an acceptable safety profile, in patients with resectable locally advanced HNSCC. Long-term survival outcomes are still pending.

## Methods

### Study design and patients

The study was approved by the Institutional Research Committee of the Sun Yat-sen University Cancer Center (No.2020-FXY-302). All participants signed the written informed consent form. This study was performed in accordance with the relevant guidelines and regulations and adhered to the ethical standards of the institutional and national research committee, as well as to the 1964 Helsinki Declaration (along with its later amendments or similar ethical standards). This study was registered at clinicaltrial.org (NCT04826679) on April 1 2021. Patients were enrolled between April 19, 2021, and March 17, 2022.

This Simon's two-stage phase II clinical trial was conducted at an integrated cancer center in China. Eligible patients were 18–70 years with histologically confirmed previously untreated, resectable, locally advanced (T2–T4, N0–N3b, M0) HNSCC of the oral cavity, oropharynx, hypopharynx, or larynx (stage III–IVb for non-oropharyngeal cancers and *HPV*-negative oropharyngeal cancer; stage II–III for *HPV*-positive oropharyngeal cancer, according to the 8th Edition of American Joint Committee on Cancer [AJCC] guideline). Other inclusion criteria were an Eastern Cooperative Oncology Group (ECOG) performance status of 0 or 1, at least one measurable lesion per the Response Evaluation Criteria in Solid Tumors (RECIST) version 1.1, and adequate organ function. Patients were excluded if they had suspect or known autoimmune disease, human immunodeficiency virus infection, hepatitis B or C virus infection, previously treated with anti-PD-1, anti-PD-L1, anti-PD-L2, or anti-CTLA4 inhibitors, or had received immunosuppressive medication. All patients were evaluated by a head and neck surgeon before enrollment. *HPV* status was determined using the fluorescent quantitative polymerase chain reaction (PCR) in primary tumors or enlarged lymph nodes.

### Procedures

Patients received neoadjuvant chemoimmunotherapy with camrelizumab (200 mg) plus nab-paclitaxel (260 mg/m$^2$) and cisplatin (60 mg/m$^2$) on day one of each 3-week cycle for three cycles. Treatment was discontinued in case of unacceptable toxic effects or disease progression. Symptomatic treatment was given during NeoCPC. Two to three weeks after completion of three cycles of NeoCPC, a global evaluation, including CT, MRI, and/or PET-CT, was performed to guide the subsequent therapy following a multidisciplinary team (surgeon, medical oncologist, and radiologist) discussion. Patients with laryngeal or hypopharyngeal cancer whose disease responded well to NeoCPC (complete response [CR] or partial response [PR] per RECIST version 1.1[62] received (chemo)radiotherapy directly or following cervical lymph node dissection to preserve organ function. All patients with oral or oropharyngeal cancer and the patients with laryngeal or hypopharyngeal cancer whose disease did not respond to NeoCPC (stable disease [SD] or progressive disease [PD]) proceeded to surgery. For patients with poor physical condition and/or heavy tumor burden unsuitable for surgery and those who refused surgery, radiotherapy alone or concurrent chemoradiotherapy was applied. Postoperative adjuvant (chemo)radiotherapy was prescribed as clinically indicated according to the National Comprehensive Cancer Network (NCCN) Clinical Practice Guidelines.

The radiographic assessment was performed at baseline, 9 weeks after treatment initiation, and every 6 months thereafter, according to the RECIST v1.1. The primary specimens and enlarged lymph nodes were obtained at baseline and at the time of surgery. Pathologic response was determined by senior pathologists who were blinded to the clinical data of the patients. Pathologic treatment effect (PTE) was defined as the area showing necrosis with associated histiocytic inflammation and/or giant cell reaction to keratinaceous debris divided by total tumor area. The pathologic response was classified as no (NPR, PTE < 20%), partial (PPR, 20% ≤PTE < 90%), major (MPR, PTE ≥ 90%), or complete (pCR, PTE = 100%) pathologic response. Both the primary tumors and lymph nodes were used to assess pathologic response. Pre- and post-treated tumor and blood samples were obtained for post hoc exploratory analyses.

Adverse events were monitored and recorded throughout the study treatment and continued for 30 days after the last treatment dose. All adverse events were graded according to the Common Terminology Criteria for Adverse Events (CTCAE) version 5.0. Postsurgical complications were classified according to the Clavien-Dindo system.

### Endpoints

The primary endpoint was the objective response rate (ORR), defined as the proportion of patients achieving CR or PR per RECIST v1.1. Secondary endpoints included pathologic complete response (pCR), major pathologic response (MPR), 2-year progression-free survival (PFS) rate, 2-year OS rate, and toxicities. The pCR was defined as a PTE of 100%, while the MPR was defined as a PTE of no less than 90% in both the primary tumors and lymph nodes. The PFS was defined as the time from the date of treatment initiation to the date of disease recurrence or death of any cause. The OS was defined as the time from the date of treatment initiation to the date of death of any cause.

### Immunohistochemical analysis

PD-L1 expression was assessed by using the 22C3 pharmDx assay (1:50, Dako, M3653) on the DAKO Autostainer Link 48 platform. The specimens with a minimum of 100 viable tumor cells were used[63]. The combined positive score (CPS) was calculated as the number of PD-L1-positive cells (tumor cells, macrophages, and lymphocytes) divided by the total number of tumor cells and multiplied by 100. A CPS of ≥1 was considered PD-L1-positive.

### Multiplex immunofluorescence

Multiplex immunofluorescence staining was performed in pre- and post-treated samples for all the patients to identify tumor infiltration immune cell subsets in the tumor microenvironment using a PANO 6-plex IHC kit (TSA-RM-82758, Panovue, Beijing, China) and PANO 7-plex IHC kit (cat 0004100100; Panovue, Beijing, China). T cells were identified by using the CD8 and CD4. Natural killer (NK) cells were identified by using the CD56 marker and were divided into weak (CD56dim) and strong (CD56bright) staining categories according to cell membrane staining intensity. Tumor-associated macrophages (TAMs) were identified by using CD68 and HLA-DR markers and were divided into M1-like (CD68+ and HLA-DR + ) and M2-like (CD68+ and HLA-DR − ) subtypes. T regulatory (Treg) cells were identified by using FoxP3 marker. Pretreated specimens were sequentially incubated with the primary antibodies (CD4 1:100 dilution; CD8 1:100 dilution; CD56 1:100 dilution; CD68 1:100 dilution; HLA-DR- 1:100 dilution; pan-CK 1:1000 dilution) and horseradish peroxidase (HRP)-conjugated secondary antibodies (1:3 dilution) and tyramide signal amplification. Nuclei were then stained with 4'−6'-diamidino-2-phenylindole (DAPI, SIGMA-ALDRICH). The number of cells from three random fields of view per sample was counted.

The stained slides were scanned using the Mantra System (PerkinElmer, Waltham, MA, USA) to obtain multispectral images. The scans captured were combined to build a single stack image.

### Image analysis

Images of single-stained sections were used to extract the spectrum of autofluorescence of each fluorescein. Those of unstained sections were used to extract the spectrum of autofluorescence of tissues. The extracted images were used to establish a spectral library that was required for multispectral unmixing by using the inForm image analysis software (PerkinElmer, Waltham, MA, USA). Reconstructed images of sections with the autofluorescence removed were generated using the spectral library.

## Targeted next-generation sequencing (NGS) and data processing

Genomic DNA from FFPE sections or biopsy samples and whole blood control samples were extracted using the QIAamp DNA FFPE Tissue kit and DNeasy Blood and Tissue Kit (Qiagen, USA), respectively. Sequencing libraries were prepared using the KAPA Hyper Prep Kit (KAPA Biosystems) according to the manufacturer's instructions for different sample types. Customized xGen Lockdown Probes (Integrated DNA Technologies) targeting 437 cancer-related genes were used for hybridization enrichment. The capture reaction was performed with Dynabeads M-270 (Life Technologies) and xGen Lockdown Hybridization and Wash Kit (Integrated DNA Technologies) according to the manufacturer's protocols. Captured libraries were constructed by on-beads PCR amplification with Illumina p5 (5'-AAT GAT ACG GCG ACC ACC GA-3') and p7 primers (5'-CAA GCA GAA GAC GGC ATA CGA GAT-3') on the KAPA HiFi HotStart ReadyMix (KAPA Biosystems), followed by purification using Agencourt AMPure XP beads. Libraries were quantified by quantitative PCR using the KAPA Library Quantification Kit (KAPA Biosystems). Library fragment size was determined by Bioanalyzer 2100 (Agilent Technologies). The target-enriched library was then sequenced on the HiSeq4000 NGS platform (Illumina) according to the manufacturer's instructions. The mean coverage depth was 317X for the whole blood control samples and 1320X for the tumor samples.

## Mutation calling

Trimmomatic was used for FASTQ file quality control to remove leading or trailing low-quality bases (quality reading <20) and N bases. Paired-end reads were aligned to the reference human genome (build hg19) using the Burrows−Wheeler Aligner. The PCR deduplication was performed by using the Picard and local realignment around indels. The base quality score was recalibrated using GATK3. Matched tumor and normal sample pairs were then checked for the same single nucleotide polymorphism (SNP) fingerprint using VCF2LR (GeneTalk), and the non-matched samples were removed. Besides, samples with mean depth <30X for blood and <500× for tissue were removed. Somatic single nucleotide variants (SNV) were called by using Mutect and Insertion/Deletions (INDELs) were called by running Scalpel (scalpel-discovery in-somatic mode). The SNVs and INDELs called were further filtered based on the following criteria: (1) a minimum of ≥5 variant supporting reads and ≥1% variant allele frequency supporting the variant; (2) present in >1% population frequency in the 1000 g or ExAC database; (3) filtered through an internally collected list of recurrent sequencing errors (≥3 variant reads and ≤20% variant allele frequency in ≥30 out of ~2000 normal samples). A final list of mutations was annotated using vcf2maf (call VEP for annotation). Panel TMB was counted by summing all base substitutions and indels in the coding region of targeted genes, including synonymous alterations to reduce sampling noise and excluding known driver mutations that are over-represented, as previously described[33].

## Statistical analysis

This study used Simon's two-stage design. The null hypothesis p0 was set at an ORR rate of 72%[35], while the alternative hypothesis was an ORR rate of 88%. Assuming a two-sided α value of 0.05 and a power (1-β) of 80%, 11 patients were required to be enrolled in the first stage. If fewer than eight patients achieved a CR or PR, the trial would be permanently stopped. Otherwise, the trial would continue to enroll an additional 36 patients in the second stage. The null hypothesis could be rejected if 39 or more patients achieved a CR or PR.

The binary endpoints (ORR and pCR) were expressed as frequency and percentage, and corresponding 95% confidence intervals (CIs) were estimated using the normal approximation method. The correlation of potential biomarkers (PD-L1, HPV infection, tumor infiltration immune cell subsets) with the radiographic and pathologic response was estimated using Spearman's rank correlation coefficient. The correlation between PD-L1 expression (CPS ≥ 1 vs CPS < 1) or radiographical response (ORR vs non-ORR) and pathologic response (pCR vs non-pCR) was evaluated using the Fisher exact test. The PFS and OS were estimated using the Kaplan−Meier method and the corresponding 95% confidence intervals (CIs) were estimated with the Log-Log transformation method. Safety was summarized in treated patients using descriptive statistics.

Oncoplots, constructed by R (3.5.2), were generated to view the overall mutation landscape of patients. Fisher exact test was used to compare the proportions between groups. A two-sided $P < 0.05$ was considered significant for all tests, unless otherwise indicated. All statistical analyses were performed by using the R V3.5.2.

## Reporting summary

Further information on research design is available in the Nature Portfolio Reporting Summary linked to this article.

## Data availability

The study protocol is available as a Supplementary Note in the Supplementary Information file. All raw targeted DNA-sequencing data generated in this study have been deposited in the National Genomics Data Center (NGDC) under the accession code HRA005542. The raw sequencing data contain information unique to individuals are available under controlled access. Access to the data can be requested by completing the application form via GSA-Human System and is granted by the corresponding Data Access Committee. Additional guidance can be found at the GSA-Human System website [https://ngdc.cncb.ac.cn/gsa-human/document/GSA-Human_Request_Guide_for_Users_us.pdf]. The raw patient data are protected and not publicly available due to data privacy laws. The de-identified individual patient data will be available upon reasonable request for academic research purposes by contacting the corresponding author (liuxk@sysucc.org.cn) for 10 years. The remaining data are available within the Article, Supplementary Information, and Source. Source data are provided with this paper.

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

## Acknowledgements

This study was supported by the China Postdoctoral Science Foundation (No.2022M713612) and the Natural Science Foundation of Guangdong Province, China (No. 2021A1515012575). The funders have no role in study design, data collection and analysis or manuscript writing.

## Author contributions

D.W., X.Z., Z.L., and X.L. took part in the conceptualization of the study; D.W. and X.L. supervised the study and were responsible for the resources and funding acquisition; D.W., Yong L., P.X., Q.F., F.C., H.L., Yin L., Y.Z.L., Z.Z., X.H., G.L., and X.L. participated in the study and collected the data. D.W., P.X., Y.T., J.S., H.Y., and Y.F. validated the data. Yong L. and P.X. performed pathology analysis. Y.S., L.L., and L.C. took part in formulating the radiotherapy program. D.W. drafted the manuscript. All authors reviewed and revised the manuscript and provided final approval.

## Competing interests

The authors declare no competing interests.

## Additional information

Di Wu [1,10], Yong Li [2,10], Pengfei Xu [1,10], Qi Fang[1], Fei Cao [1], Hongsheng Lin[1], Yin Li[1], Yong Su[3], Lixia Lu[3], Lei Chen[3], Yizhuo Li[4], Zheng zhao[1], Xiaoyu Hong[5], Guohong Li[5], Yaru Tian [6], Jinyun Sun[6], Honghong Yan[7], Yunyun Fan[1], Xinrui Zhang [8,11] ✉, Zhiming Li [9,11] ✉ & Xuekui Liu [1,11] ✉

[1]Department of Head and Neck Surgery, Sun Yat-Sen University Cancer Centre, State Key Laboratory of Oncology in South China, Collaborative Innovation Center for Cancer Medicine, Guangzhou, China. [2]Department of Pathology, Sun Yat-Sen University Cancer Centre, State Key Laboratory of Oncology in South China, Collaborative Innovation Center for Cancer Medicine, Guangzhou, China. [3]Department of Radiation Oncology, Sun Yat-Sen University Cancer Centre, State Key Laboratory of Oncology in South China, Collaborative Innovation Center for Cancer Medicin, Guangzhou, China. [4]Department of Radiology, Sun Yat-Sen University Cancer Centre, State Key Laboratory of Oncology in South China, Collaborative Innovation Center for Cancer Medicine, Guangzhou, China. [5]Nanjing Geneseeq Technology Inc, Nanjing, China. [6]Jiangsu Hengrui Pharmaceuticals Co., LTD, Shanghai, China. [7]Department of Intensive Care Medicine, Sun Yat-Sen University Cancer Centre, State Key Laboratory of Oncology in South China, Collaborative Innovation Center for Cancer Medicine, Guangzhou, China. [8]Department of Otolaryngology-Head and Neck Surgery, The Fifth Affiliated Hospital of Guangzhou Medical University, Guangzhou, China. [9]Department of Medical Oncology, Sun Yat-Sen University Cancer Centre, State Key Laboratory of Oncology in South China, Collaborative Innovation Center for Cancer Medicine, Guangzhou, China. [10]These authors contributed equally: Di Wu, Yong Li, Pengfei Xu. [11]These authors jointly supervised this work: Xinrui Zhang, Zhiming Li, Xuekui Liu. ✉e-mail: zxr-850@163.com; lizhm@sysucc.org.cn; liuxk@sysucc.org.cn

