## [Peer Review File · Nature Communications]

Neoadjuvant Chemoimmunotherapy with Camrelizumab plus Nab-paclitaxel and Cisplatin (NeoCPC) in Resectable Locally Advanced Squamous Cell Carcinoma of the Head and Neck: A Pilot Phase II TrialREVIEWER COMMENTS

Reviewer #1 (Remarks to the Author): with expertise in HNSCC, immunotherapy

In this single arm phase 2 study of neoadjuvant chemo-immunotherapy for HNSCC, the authors use a combination of PD1-blockade combined with two chemotherapy agents (taxol plus cisplatin) in a cohort of 48 patients. Of this group, 27 went to on have surgery while the remainder were treated non-surgically. Biomarker discovery was based on correlation of PDL1 staining, mutations status (NGS panel sequencing of 400+ genes) and limited multiplex IHC looking for CD8 T cells, NK cells and TAMs. They showed a remarkable ORR of >80% with a number of patients achieving complete radiological or pathological response. These were biased to HOPV positive cases, and a number of other factors- TP53 and TERT mutations, CD8 density and M1 macrophage counts. Unfortunately the success of this regime mean that comparisons could mainly be done on patients with significant response versus those who had stable disease at worst, and hence there is a lack extreme non-responders to act as a good control. This is not a criticism of the study team, but just the state of play, and hence their comparisons all end up being a bit weak for that reason.

The trial is an excellent study, and I am really impressed that all 48 patients completed the relatively harsh regime applied here, and the many went on receive definite treatment after that. The most remarkable finding here is that HPV positive patients has the best response to this combination of cytotoxic plus PD1 blockade. It is likely that this same HPV positive population is also TP53 and TERT mutation negative and hence may bias the latter association. Similarly, the high CD8 and M1 macrophage TME is also correlated with HPV, and all in all, the biomarkers suggested here may simply be driven by HPV positivity, unless the authors can demonstrate these associations in only HPV negative tumors.

I have a number of further queries issues listed below that need to be addressed:

1. the standard consort diagram and trial diagram is needed and the former should be included as a main figure rather than in supplementary data.
2. I find the order in which the data is presented to be a bit strange. I feel that the radiographic data and figures 4 and 5 should come before 2C,D,E and figure 3, so that you explain some of the mutation and mIHC findings in greater detail before showing correlation

with pathologic response.

3. Is the data sufficiently mature to show survival data now since it is now 18 months since the end of the study.

4. I suggest showing dot plot of TPS and CPS score with response to show trends, or categories of PDL1 Scores (eg 1-10, 10-20 and > 20 CPS and TPS) for both radiological and pathologic response

5. Are there correlations between TP53 and TERT mutations with HPV (as would be expected). Also, any differences between TP53- point versus deletion mutants?

6. Is the histological correlation between HPV and higher cd8/M1 macrophage the reason for the association of latter with response? p53 and TERT is also biased to HPV positive cases, so stats are biased- can u see a difference if Figures 5F and 5G are done for HPV negative cases only

7. The mIHC is extremely limited to 3 cell populations which is rather incomplete. I feel that there is also a need to evaluate CD4 and Treg populations, but also deeper profiling of exhaustion markers etc. These are especially relevant when comparing pre- and post-treatment samples and ask what are the TME effects on these tumors after Neo-adjuvant Rx.

7. Is the change after treatment of M1 macrophages seen in all cases or only responders (vs non responder?)

8. In the discussion section, the response rates for pembro, nivo and nivo plus ipi seems inaccurate if you consider all the trials and existing data, can you please check the different studies on this.

Reviewer #2 (Remarks to the Author): with expertise in HNSCC, immunotherapy

Induction chemotherapy is not standard of care for head and neck cancer. The "standard" TPF regimen was established via 2 trials showing superiority of TPF over PF, yet neither was properly studied compared to standard of care alone, and neither administered properly standard definitive therapy. Further, the regimen is toxic and requires an infusion port. This has led to extensive studies of alternative regimens. Most accepted of these may be the Kies regimen of carbo/taxol/cetuximab. This history and the reasons for seeking an alternative to TPF deserve further mention. Additionally, Weiss and Adkins have studied the role of

abraxane in induction and background surrounding abraxane was not discussed.

The study itself demonstrated impressive results for clinical response and pathologic response. Given that camrelizumab is a newer agent, there is a path to phase III study. The correlative results, while they will not inform practice, are interesting.

It is unclear why the discussion section reviews induction in breast cancer when this literature exists in head/neck cancer. At least two studies have shown absence of a relationship between clinical and pathologic response in head/neck cancer, an interesting contrast to these results.

Reviewer #3 (Remarks to the Author): with expertise in clinical trial study design, biostatistics

1. The variance estimate used for the confidence interval of some proportions seems quite small compared to the total number of patients used for calculating the corresponding proportions, e.g., for the major pathologic response 63% is computed based on 27 patients, was the variance estimate consistent with this? This seems also the case for pathologic complete response, and clinical to pathological downstaging.

2. There are some small numerical inconsistencies between tables/figures and text:

a. in Table S1, 23 patients are reported to receive adjuvant chemo or radiotherapy, while in the text it is reported that overall 25 patients received it.

b. Also, in Table 2, only 2 Grade 3-4 TRAEs are reported, and only 1 peripheral neuropathy, but the text refers to three grade 3 or worse TRAEs.

c. In Figure 2B outcomes for 27 patients are reported but the text states that focus would be on 26.

3. Figure 2B also tests for association between pathologic and radiographic response. I infer from the text that SDs are deterministically assigned to Non-pCR, and CRs to pCR. If this is the case, some association would be present by construction. Could you perhaps expand on the adopted test and hypotheses?

4. Although standard classifications are used, I would suggest to introduce all abbreviations the first time they are mentioned in the text, as well as how classes are aggregated. For

example, there is some slight ambiguity on when MPR also includes pCR, and Non-ORR is only mentioned to include SD but not PD in the text.

5. In general, further clarifications about the adopted statistical tests should be provided for all comparisons (Figure 2 D-E, Figure 3, Figure 4 B-C, and Figure 5 E-G, Figure S6). Note that Figure 5G reports no significance indication.

6. The definition of 'Middle' and 'High' in Figure 1 can be inferred from the text, but is not explicitly given. It would be helpful to explicitly associate them with the CPS values.

7. The text mentions that genetic differences between pCR and Non-pCR are tested and found non-significant, could you add a short description of the testing approach used?

8. It may be a problem with the rendering, but in Figure 4A pink and blue appear in the plot but not in the legend.

9. It is mentioned that in Figure S4 no distinguishing distribution pattern was identified: also in this case I would report the test, or that this was inferred from visual inspection.

10. Figure S5 resolution is quite low.

11. This does not affect the results, but in the adopted Simon's two-stage design the null hypothesis can be rejected if 39 or more patients achieved a CR or PR.

Point-by-point response to the reviewers' comments

Response to Reviewers' comments:

Reviewer #1 (Remarks to the Author): with expertise in HNSCC, immunotherapy

In this single arm phase 2 study of neoadjuvant chemo-immunotherapy for HNSCC, the authors use a combination of PD1-blockade combined with two chemotherapy agents (taxol plus cisplatin) in a cohort of 48 patients. Of this group, 27 went to on have surgery while the remainder were treated non-surgically. Biomarker discovery was based on correlation of PDL1 staining, mutations status (NGS panel sequencing of 400+ genes) and limited multiplex IHC looking for CD8 T cells, NK cells and TAMs. They showed a remarkable ORR of >80% with a number of patients achieving complete radiological or pathological response. These were biased to HOPV positive cases, and a number of other factors- TP53 and TERT mutations, CD8 density and M1

macrophage counts. Unfortunately the success of this regime mean that comparisons could mainly be done on patients with significant response versus those who had stable disease at worst, and hence there is a lack extreme non-responders to act as a good control. This is not a criticism of the study team, but just the state of play, and hence their comparisons all end up being a bit weak for that reason.

Response: Thank you for your comments. Indeed, our study showed the promising anti-tumor activity of neoadjuvant therapy with camrelizumab plus nab-paclitaxel and cisplatin, leaving few patients with disease progression. Accordingly, patients who had stable disease were primarily used as control group for between group comparison, which might be a bit weak. We have worked on a phase III randomized trial with a larger number of patients to provide additional data sources for us to explore the results further. We have added some limitations in the discussion section, as shown below.

“Also, exploratory analysis was done on patients with ORR versus those who had non-ORR, yet only five patients were included in the non-ORR group (four SDs and one PD), which may make the comparisons a bit weak, future studies are needed to determine the prognostic values of PD-L1 CPS and other biomarkers.” (**Discussion section: Page 20 Lines 420-424**)

The trial is an excellent study, and I am really impressed that all 48 patients completed the relatively harsh regime applied here, and the many went on receive definite treatment after that. The most remarkable finding here is that HPV positive patients has the best response to this combination of cytotoxic plus PD1 blockade. It is likely that this same HPV positive population is also TP53 and TERT mutation negative and hence may bias the latter association. Similarly, the high CD8 and M1 macrophage TME is also correlated with HPV, and all in all, the biomarkers suggested here may simply be driven by HPV positivity, unless the authors can demonstrate these associations in only HPV negative tumors.

Response: Thank you for your constructive comments. We have performed correlation analyses of TP53 and TERT mutations with HPV status according to your suggestions. As shown in Figure 1 below (Figure 4E in the main text), the TP53 mutation was significantly lower in HPV+ patients than in HPV- ones ($p < 0.001$), while the TERT mutation was relatively lower in HPV+ patients, although the difference did not reach statistical significance ($p = 0.09$). Given the limited number of HPV+ patients ($n = 9$), further correlation analysis of TP53/TERT mutation with radiographical response was performed in the HPV-negative patients. Among the HPV-negative patients, there were no significant differences in radiographic tumor response regarding TP53 or TERT mutation (Figures 1F and 1G). Consequently, we could not preclude the possibility that the same HPV+ population is also TP53 and TERT mutation negative which may bias the latter association. We have added the above-mentioned data in the results and discussion sections. As for the CD8 and M1 macrophage TME, we have investigated their correlations with radiographical tumor response in HPV-positive and HPV-negative patients. No significant correlations of CD8+ T cells or M1 macrophage cells were observed (Figure 2), suggesting that their

radiographical response correlations may be biased to some extent by HPV mutation of the patients. We have added corresponding information in the results and discussion sections.

Figure 1 The genetic landscape and radiographic tumor response analysis. Alterations as assessed by next-generation sequencing of baseline primary tumor samples (A); Comparisons of radiographic tumor response between HPV-positive and HPV-negative patients (B); TP53-mutant and TP53-wildtype patients (C); and TERT-mutant and TERT-wildtype patients by using the unpaired t test with Welch's correction (D). Correlation between HPV status and TP53 or TERT mutations analyzed by using the unpaired t test with Welch's correction (E). Comparisons of radiographic tumor response between TP53-mutant and TP53-wildtype patients (F) and TERT-mutant and

TERT-wildtype patients (G) in HPV-negative patients by using the unpaired t test with Welch's correction. *: $p < 0.5$, **: $p < 0.01$, ***: $p < 0.001$, ns: No significance.”

Figure 2 Correlation analyses of radiographic tumor response with the densities of tumor-infiltrating CD8+ T cells (A and C) and tumor-infiltrating M1 macrophage cells (B and D) in HPV-positive (A and B) and HPV-negative patients (C and D).

“The correlation analyses of TP53/TERT alternations with HPV were further performed. As shown in Figure 4E, the TP53 alternation was more common in HPV- patients ($p < 0.001$), while the TERT alternation was also more common in HPV- patients, although the difference did not reach statistical significance ($p = 0.09$). The correlations of radiographical tumor response with TP53/TERT alternations were further carried out among the HPV- patients. There were no significant differences in radiographic response regarding TP53 and TERT alternation (Figure 4F and 4G).” **(Results sections: Page 11 Lines 232-Page 12 Line 238)**

“Further exploratory analysis showed no significant correlations between CD8+ T cells or M1-like macrophages and radiographical tumor response in HPV+ and HPV- patients.” **(Results sections: Page 12 Lines 248-250)**

“Interestingly, TP53 and TERT alternations were found to be enriched in HPV-negative patients, while more intratumor CD8+ T cells and M1-like macrophage cells were found in HPV-positive patients. That is, the same HPV-positive population is also TP53/TERT wildtype, while has high levels of CD8 and M1-like macrophage tumor microenvironment. To better understand the potential role of HPV in response correlation of the biomarkers, further correlation analysis of radiographical response with TP53/TERT alternation was performed in the HPV-negative and/or HPV-positive populations. There were no significant differences in radiographic response regarding TP53 or TERT alternation in the HPV-negative patients, or intratumor CD8+ T cells and M1-like macrophage cells in HPV-positive and HPV-negative patients, suggesting that their radiographical response correlations may be biased to some extent by HPV mutation of patients. Unfortunately, the sample size is small in this study, for example, there were only nine HPV-positive patients and three HPV-negative patients being TP53 wildtype, which precluded the possibility for us to explore the potential correlation of TP53/TERT alternation, immune infiltrating cells, and HPV status in HNSCC. Whether the correlations of these biomarkers with radiographic response was biased by HPV status of patients still needs further investigation.” **(Discussion sections: Page 19 Lines 398-Page 20 Line 413)**

I have a number of further queries issues listed below that need to be addressed:

1. the standard consort diagram and trial diagram is needed and the former should be included as a main figure rather than in supplementary data.

Response: We have included the trial diagram as Figure 1 in the main text according to your suggestions.

2. I find the order in which the data is presented to be a bit strange. I feel that the radiographic data and figures 4 and 5 should come before 2C, D, E and figure 3, so that you explain some of the mutation and mlHC findings in greater detail before showing correlation with pathologic response.

Response: We have reordered the figures and put the Figure 4 and 5 in front of Figure 2C, D, E and figure 3 (as Figure 6) according to your suggestions.

3. is the data sufficiently mature to show survival data now since it is now 18 months since the end of the study.

Response: Thank you for your question. The survival data were still not mature at the data cutoff (August 30, 2023), as shown in the Figure 3 below (not mentioned in the manuscript). The median follow-up was 666 days (95% CI: 537, 753). Four patients had disease progression and one had died. We have updated the above-mentioned data in the results section.

“As of the data cutoff (August 30, 2023), the median follow-up time was 666 (IQR: 537, 753) days. Survival data were not mature. The estimated one-year OS and PFS rates were both 97.9% (95% CI: 94.0%, 100.0%) (Figure S2).” **(Results sections: Page 9 Lines 170-172)**

Figure 3 Kaplan-Meier analysis of progression-free survival and overall survival.

4. I suggest showing dot plot of TPS and CPS score with response to show trends, or categories of PDL1 Scores (eg 1-10, 10-20 and > 20 CPS and TPS) for both radiological and pathologic response

Response: Thank you for your suggestions. We have provided dot plots of PD-L1 CPS score with radiographical and pathological responses as shown in Figure 4 below. We prefer to not mention the figures in the manuscript to ensure the style consistency and hope it will still meet your requirement.

Figure 4 Comparison of PD-L1 expression with radiographical response and pathological response. CPS: Combined positive score; pCR: Pathological complete response; ORR: Objective response rate.

5. are there correlations between TP53 and TERT mutations with HPV (as would be expected). Also, any differences between TP53- point versus deletion mutants?

Response: Thank you for your suggestion. We have provided correlation analyses of TP53/TERT mutations with HPV status and added the above-mentioned data in the results section, as mentioned above. Besides, we have further analyzed the correlations between TP53- point versus deletion mutants and HPV status. There was no difference between TP53- point versus deletion mutants with respect to HPV status, as shown in Table 1 below.

Table 1 Correlation between TP53 mutation type and HPV.

	HPV+ (n=2)	HPV- (n=35)	p value
TP53 mutation type, n (%)			1.000
POINT	2 (100.0)	25 (71.4)	
DEL	0	8 (22.9)	
MIXED	0	2 (5.7)	

“The correlation analyses of TP53 and TERT mutations with HPV were further performed. As shown in Figure 4E, the TP53 mutation rate was significantly lower in HPV+ patients than in HPV- ones ($p < 0.001$), while the TERT mutation rate was numerically lower in HPV+ patients, although the difference did not reach statistical significance ($p = 0.09$).” (**Results sections: Page 11 Lines 232-Page 12 Line 238**)

6. Is the histological correlation between HPV and higher cd8/M1 macrophage the reason for the association of latter with response? p53 and TERT is also biased to HPV positive cases, so stats are biased- can u see a difference if Figures 5F and 5G are done for HPV negative cases only.

Response: Thank you for your suggestion. We have analyzed the correlations of CD8+ T cells and M1 macrophage cells with radiographical tumor response in HPV+ and HPV- patients. No significant differences were observed, suggesting that their radiographical response correlations may be biased to some extent by HPV mutation of the patients. We have added some information in the results and discussion sections, as mentioned above. Besides, we have also tried to analyze the correlations between CD8/M1 macrophage and TP53/TERT in HPV-negative patients. Unfortunately, only three HPV-negative patients had wildtype TP53, with no significant differences between TP53 wildtype and alternated groups, as shown in Figure 5A below. Meanwhile, there were no significant differences in the densities of CD8+ T cells and M1 macrophage cells with respect to TERT mutation in HPV-negative patients (Figure 5B). Due to the limited number of patients with mutations, we prefer not to include the results in the manuscript and hope it will meet your requirement.

“Further exploratory analysis showed no significant correlations between CD8+ T cells or M1-like macrophages and radiographical tumor response in HPV-positive and HPV-patients.” (**Results sections: Page 12 Lines 248-250**)

Figure 5 Comparisons of the densities of tumor infiltration immune cells between TP53-mutant and TP53-wildtype (A) and TERT-mutant and TERT-wildtype (B) in HPV-negative patients.

7. The mIHC is extremely limited to 3 cell populations which is rather incomplete. I feel that there is also a need to evaluate CD4 and Treg populations, but also deeper profiling of exhaustion markers etc. These are especially relevant when comparing pre- and post-treatment samples and ask what are the TME effects on these tumors after Neo-adjuvant Rx

Response: Thank you for your suggestions. We have further evaluated CD4 and Treg populations and added the correlation analysis of CD4 and Treg populations with radiographical response in Figures 6A and Figure 7L, 7I, and 7M (Figure S6A and

Figure S4L, S4I, and S4M in the revised manuscript) below. The representative images of multiplex immunofluorescence of CD4, Foxp3, and Pan-CK are shown in Figure 6B below (Figure S6B in the revised manuscript). However, we failed to evaluate deeper profiling of exhaustion markers due to the insufficient tissue samples, particularly for patients with hypopharyngeal or laryngeal cancer. We have added some limitations in the discussion section and hope it will still meet your requirements.

“The density of CD8+ T cells in the tumor area was significantly associated with the percentage of radiographic response ($p=0.0005$, Figure 5A), and so were the densities of M1-like macrophage cells ($p=0.009$, Figure 5B) and CD4+ T cells ($p=0.021$, Figure S6A), respectively. However, no significant association was found in the other tumor-infiltrating immune cell subsets, including M2-like macrophage cells, CD56bright NK cells, and CD56dim NK cells, and T regulatory (Treg) cells (Figure S4). The representative images of multiplex immunofluorescence are shown in Figures 5C, 5D, and S6B.” (Results section: Page 12 Lines 239-245)

“Multiplex immunofluorescence staining was performed to identify tumor infiltration immune cell subsets in the tumor microenvironment using a PANO 6-plex IHC kit (TSA-RM-82758, Panovue, Beijing, China) and PANO 7-plex IHC kit (cat 0004100100; Panovue, Beijing, China). T cells were identified by using the CD8 and CD4.” (Methods section: Page 24 Lines 504-507)

“T regulatory (Treg) cells were identified by using FoxP3 marker.” (Methods section: Page 24 Lines 511-512)

“Additionally, multiplex immunofluorescence was limited to only a few cell populations (CD8, M1 macrophage, CD4, and Treg) due to the insufficient tissue samples, particularly for patients with hypopharyngeal or laryngeal cancer. Future studies need to evaluate deeper profiling of exhaustion markers and their pre-and-post changes, so as to better understand the potential tumor microenvironment effects after neoadjuvant chemioimmunotherapy.” (Discussion section: Page 20 Lines 424-429)

Figure 6 Tumor-infiltrating lymphocyte analysis. Correlation analysis of radiographic tumor response with the densities of tumor-infiltrating CD4+ T cells by using the

Spearman correlation text (A). The multiplex immunofluorescence images of the tumor sites in patient 17 (B). Primary antibodies targeting CD4, FoxP3, and Pan-CK were used.

Figure 7 Correlation analysis between radiographic tumor regression and TMB, PD-L1 expression, or density of tumor-infiltrating immune cells by using the Spearman correlation test.

8. Is the change after treatment of M1 macrophages seen in all cases or only responders (vs non responder?)

Response: Thank you for your suggestions. The change in M1 macrophages after treatment is shown in all cases, including both responders and non-responders. We have added the corresponding results in responders and non-responders, as shown in Figure 8 below (Figure S5B and S5C in the revised manuscript).

Figure 8 Comparisons of the density of tumor infiltration immune cells before and after neoadjuvant therapy in overall patients (A) and those with pCR (B) and non-pCR (C) using the Mann-Whitney U. *: p<0.5, ns: No significance.”

“M1-like macrophage cells in the tumor stroma were significantly increased after treatment in the responders, but not in the non-responders, whereas the other types of tumor infiltration immune cells were not significantly changed (Figure S5).” **(Results sections: Page 13 Lines 264-266)**

9. In the discussion section, the response rates for pembro, nivo and nivo plus ipi seems inaccurate if you consider all the trials and existing data, can you please check the different studies on this.A

Response: Thank you for your suggestions. We have checked the different studies on the response rates for pembro, nivo, and nivo plus ipi, and made some corrections accordingly, as shown below.

“The pCR MPR or pCR rate was 5%~7% for pembrolizumab alone (43, 44), 5.9%~17% for nivolumab alone, and 20~35% for nivolumab combined with ipilimumab (22, 45, 46).” **(Discussion sections: Page 15 Lines 311-313)**

Reviewer #2 (Remarks to the Author): with expertise in HNSCC, immunotherapy

Induction chemotherapy is not standard of care for head and neck cancer. The "standard" TPF regimen was established via 2 trials showing superiority of TPF over PF, yet neither was properly studied compared to standard of care alone, and neither administered properly standard definitive therapy. Further, the regimen is toxic and requires an infusion port. This has led to extensive studies of alternative regimens. Most accepted of these may be the Kies regimen of carbo/taxol/cetuximab. This history and the reasons for seeking an alternative to TPF deserve further mention. Additionally, Weiss and Adkins have studied the role of abraxane in induction and background surrounding abraxane was not discussed.

Response: Thank you for your constructive comments and suggestions. We totally agree with your point about the induction chemotherapy. We have added some information about the history and reasons for seeking an alternative to TPF and alternative regimens studied in the introduction section. Besides, we have added some discussion of the role of abraxane in induction and background surrounding abraxane.

“Nab-paclitaxel was used in this study as the part of the chemotherapy regimen, instead of paclitaxel. Induction chemotherapy with nab-paclitaxel-based regimen has been reported to be associated with better survival outcomes when compared with docetaxel-based regimen (38). The potential mechanism may involve secreted protein acidic and rich in cysteine (SPARC) that plays a role in albumin receptor-mediated endometrial transport (39) and activated RAS and/or PI3K pathways that are common drivers for initiating micropinocytosis, the process by which macromolecules such as albumin are taken up into cells. SPARC expression was found in tumor and stromal cells but not in the adjacent normal oral mucosa in HNSCC (40) and correlated with

better tumor response to nab-paclitaxel in HNSCC patients (41). Meanwhile, the frequently activated RAS and/or components of the PI3K pathways may also help to explain the enhanced anti-tumor effect of nab-paclitaxel in HNSCC (42). Collectively, these data suggest that nab-paclitaxel may be a potential option in induction chemotherapy of HNSCC. Future studies are needed to clarify it further.” (Discussion sections: Page 14 Line 295-Page 15 Line 307)

The study itself demonstrated impressive results for clinical response and pathologic response. Given that camrelizumab is a newer agent, there is a path to phase III study. The correlative results, while they will not inform practice, are interesting.

Response: Thank you for your positive comments. We have already worked on the phase III study to further investigate the efficacy and safety of neoadjuvant camrelizumab plus chemotherapy, as well as potential biomarkers for prognostic prediction.

It is unclear why the discussion section reviews induction in breast cancer when this literature exists in head/neck cancer. At least two studies have shown absence of a relationship between clinical and pathologic response in head/neck cancer, an interesting contrast to these results.

Response: Thank you for your suggestions. We have deleted the studies of breast cancer in the discussion section. Meanwhile, we have added some discussion about the correlation between clinical and pathologic response, as well as their prognostic values in head/neck cancer.

“The correlations between pathological remission after neoadjuvant chemoimmunotherapy and radiographical response, and as well as their prognostic values have not been conclusively concluded. The findings of our study showed a significant correlation between radiographical response and pathological response, consistent with the previous results of neoadjuvant nivolumab (47). However, the conflicting results of no significant correlation were also reported in patients with resectable HNSCC receiving neoadjuvant chemoimmunotherapy (34, 48).” (Discussion sections: Page 15 Line 323-Page 16 Line 329)

Reviewer #3 (Remarks to the Author): with expertise in clinical trial study design, biostatistics

1. The variance estimate used for the confidence interval of some proportions seems quite small compared to the total number of patients used for calculating the corresponding proportions, e.g., for the major pathologic response 63% is computed based on 27 patients, was the variance estimate consistent with this? This seems also the case for pathologic complete response, and clinical to pathological downstaging.

Response: Thank you for your suggestions. A total of 48 patients were enrolled and

treated with neoadjuvant camrelizumab plus nab-paclitaxel and cisplatin. Of the 48 patients, 27 proceeded to surgery following the neoadjuvant therapy, while the remaining 21 received adjuvant therapy without surgery. The pathological response was evaluable only for the patients who underwent surgical resection (n=27), that is, the pathological results, like pCR, MPR, and clinical to pathological downstaging, were all evaluated in the surgical population. We have made some revision of our statement to make it easier to follow.

“Of the 27 resected patients, 17 (63.0%, 95% CI: 49.3, 76.7) achieved an MPR or pCR, with a pCR rate of 55.6% (95% CI: 41.5, 69.7) (including seven oral cancer, six HPV-positive oropharyngeal cancer, one HPV-negative oropharyngeal cancer, and one laryngeal cancer). Twenty-one out of the 27 patients had a clinical to pathological downstaging (77.8%, 95% CI: 66.0%, 89.6%).” (Results section: Page 8 Lines 156-160)

2. There are some small numerical inconsistencies between tables/figures and text:
- in Table S1, 23 patients are reported to receive adjuvant chemo or radiotherapy, while in the text it is reported that overall 25 patients received it.

Response: Thank you for your kind reminder and sorry for our mistake. We have double-checked and confirmed that 25 patients received postoperative adjuvant chemo or radiotherapy in this study. we have made some correction about the Figure S1 and put it as Figure 9 below (Figure 1 in the revised manuscript).

Figure 9 Study flow chart. 48 patients were enrolled in this trial and received neoadjuvant therapy; among them, 21 patients received adjuvant therapy without surgery, 27 patients underwent surgical resection.

- Also, in Table 2, only 2 Grade 3-4 TRAEs are reported, and only 1 peripheral neuropathy, but the text refers to three grade 3 or worse TRAEs

Response: Thank you for your comments and sorry for our mistake. We have double-checked and confirmed that two patients experienced TRAEs of grade 3-4, including one with peripheral neuropathy and another with pneumonitis, as shown in Table 2. We have made the corresponding revision in the abstract and results section.

“Treatment-related adverse events of grade 3 or 4 occurred in two patients.” (**Abstract: Page 3 Line 50**)

“TRAEs of grade 3 or worse occurred in two (6.3%) patients. One patient experienced grade 3 peripheral neuropathy with lower extremity numbness, mobility problems, and difficulty walking. After neurotrophic therapy, the patient gradually recovered within three months after NeoCPC.” (**Results section: Page 9 Lines 177-180**)

c. In Figure 2B outcomes for 27 patients are reported but the text states that focus would be on 26.

Response: Thank you for your suggestion. It should be 26 patients in the (new) Figure 3B. One patient with oral cancer (Patient 19) refused surgery for personal and economic reasons after completing NeoCPC and returned to the hospital for surgery 191 days later. The patient was excluded from the correlation analysis between radiographic response and pathologic response. We have made the corresponding revision, as shown in the Figure 10 below (Figure 3 in the main text).

Figure 10 Correlation analysis of radiographic response and pathologic response. Sankey plot shows the relationship between radiographic response and pathologic response (A). The consistency between radiographic response and pathologic response analyzed by using the Fisher exact test (B).

a. in Table S1, 23 patients are reported to receive adjuvant chemo or radiotherapy, while in the text it is reported that overall 25 patients received it.

Response: Thank you for your comments and sorry for our mistake. We have double-checked and confirmed that 25 patients received postoperative adjuvant chemo or radiotherapy in this study. we have made some correction about the Figure S1 and put it as Figure 1 in the revised manuscript as mentioned above.

b. Also, in Table 2, only 2 Grade 3-4 TRAEs are reported, and only 1 peripheral neuropathy, but the text refers to three grade 3 or worse TRAEs

Response: Thank you for your comments and sorry for our mistake. We have double-

checked and confirmed that two patients experienced TRAEs of grade 3-4, including one with peripheral neuropathy and another with pneumonitis, as shown in Table 2. We have made the corresponding revision of the description in the results section.

“TRAEs of grade 3 or worse occurred in two (6.3%) patients. One patient experienced grade 3 peripheral neuropathy with lower extremity numbness, mobility problems, and difficulty walking. After neurotrophic therapy, the patient gradually recovered within three months after NeoCPC.” (**Results section: Page 9 Lines 177-180**)

c. In Figure 2B outcomes for 27 patients are reported but the text states that focus would be on 26.

Response: Thank you for your kind reminder. It should be 26 patients in the (new) Figure 3B. We have made the corresponding revision, as mentioned above.

3. Figure 2B also tests for association between pathologic and radiographic response. I infer from the text that SDs are deterministically assigned to Non-pCR, and CRs to pCR. If this is the case, some association would be present by construction. Could you perhaps expand on the adopted test and hypotheses?

Response: Thank you for your suggestion. The correlation between radiographical response and pathologic response was evaluated by using the Fisher exact test. We have added some description in the statistical analysis of the methods section and figure legend.

“The correlation between PD-L1 expression (CPS \geq 1 vs CPS $<$ 1) or radiographical response (ORR vs non-ORR) and pathologic response (pCR vs non-pCR) was evaluated using the Fisher exact test.” (**Methods section: Page 27 Lines 575-577**)

“**Figure 3** Correlation analysis of radiographic response and pathologic response. Sankey plot shows the relationship between radiographic response and pathologic response (A). The consistency between radiographic response and pathologic response analyzed by using the Fisher exact test (B).”

4. Although standard classifications are used, I would suggest to introduce all abbreviations the first time they are mentioned in the text, as well as how classes are aggregated. For example, there is some slight ambiguity on when MPR also includes pCR, and Non-ORR is only mentioned to include SD but not PD in the text.

Response: Thank you for your suggestion. We have double-checked and made some correction throughout the manuscript, as shown below.

“Of the 27 patients who underwent surgery, 17 (63.0%, 95% CI: 49.3, 76.7) achieved a major or complete pathologic response, with a pathologic complete response rate of (55.6%, % (95% CI: 41.5, 69.7).” (**Abstract sections: Page 3 Lines 47-50**)

“Of the 27 resected patients, 17 (63.0%, 95% CI: 49.3, 76.7) achieved an MPR or

pCR, with a pCR rate of 55.6% (95% CI: 41.5, 69.7)” (Results section: Page 8 Lines 156-158)

“Three cycles of NeoCPC induced a major or complete pathologic response rate of 63.0% (17/27) in surgical patients, with a pCR rate of 55.6% (15/27)” (Discussion section: Page 15 Lines 308-309)

“No significant difference was observed in the PD-L1 expression, HPV status, somatic variants of 437 genes, and TMB between ORR (CR+PR) and Non-ORR (SD+PD) groups. Similarly, no significant difference was noted between CR and Non-CR (PR+SD+PD) groups.” (Results section: Page 11 Lines 225- 228)

“We also detected the alterations in these genes in this study, but did not find any difference between patients with ORR (CR+PR) and Non-ORR(SD+PD), with CR and Non-CR (PR+SD+PD), as well as those with pCR and Non-pCR.” (Discussion section: Page 18 Line 389-Page 189 Line 392)

5. In general, further clarifications about the adopted statistical tests should be provided for all comparisons (Figure 2 D-E, Figure 3, Figure 4 B-C, and Figure 5 E-G, Figure S6). Note that Figure 5G reports no significance indication.

Response: Thank you for your suggestions. We have added the statistical tests used for all comparisons in the Figure legends, as shown below.

“**Figure 3** Correlation analysis of radiographic response and pathologic response. Sankey plot shows the relationship between radiographic response and pathologic response (A). The consistency between radiographic response and pathologic response analyzed by using the Fisher exact test (B).”

“**Figure 4** The genetic landscape and radiographic tumor response analysis. Alterations as assessed by next-generation sequencing of baseline primary tumor samples (A); Comparisons of radiographic tumor response between HPV-positive and HPV-negative patients (B); TP53-mutant and TP53-wildtype patients (C); and TERT-mutant and TERT-wildtype patients by using the unpaired t test with Welch's correction (D). Correlation between HPV status and TP53 or TERT mutations analyzed by using the unpaired t test with Welch's correction (E). Comparisons of radiographic tumor response between TP53-mutant and TP53-wildtype patients (F); and TERT-mutant and TERT-wildtype patients (G) in HPV- patients by using the unpaired t test with Welch's correction. *: $p < 0.5$, **: $p < 0.01$, ***: $p < 0.001$, ns: No significance.”

“**Figure 5** Tumor-infiltrating lymphocyte analysis. Correlation analysis of radiographic tumor response with the densities of tumor-infiltrating CD8+ T cells (A) and tumor-infiltrating M1 macrophage cells by using the Spearman correlation text (B). The multiplex immunofluorescence images of the tumor sites in patient 20 (C) and patient 45 (D). Primary antibodies targeting CD8, CD56, CD68, HLA-DR and Pan-CK were

used. Comparisons of the densities of tumor-infiltrating CD8+ cells and M1 macrophage cells between HPV-positive and HPV-negative patients (E); TP53-mutant and TP53-wildtype patients (F), and TERT-mutant and TERT-wildtype patients (G) analyzed by using the unpaired t test. *: $p < 0.5$, **: $p < 0.01$, ***: $p < 0.001$, ns: No significance.”

“**Figure 6** Exploratory analysis of pathologic response characteristics. The percentages of patients with HPV infection (A), PD-L1 positive expression (B), or TP53 mutation in the pCR and non-pCR patients (C). Comparisons are made by using the Fisher exact test. Comparison of the density of tumor-infiltrating immune cells in patients with pCR and non-pCR by using the Wilcoxon rank sum test (D). ns: No significance.”

“**Figure S3** The distribution of TMB and PD-L1 expression level in patients with different anatomic sites and TNM stages analyzed by using the Wilcoxon rank sum test. ns: No significance.”

“**Figure S4** Correlation analysis between radiographic tumor regression and TMB, PD-L1 expression, or density of tumor-infiltrating immune cells by using the Spearman correlation test.”

“**Figure S5** Comparisons of the density of tumor infiltration immune cells before and after neoadjuvant therapy in overall patients (A) and those with pCR (B) and non-pCR (C) using the Mann-Whitney U. *: $p < 0.5$, ns: No significance.”

“**Figure S6** Correlation analysis of radiographic tumor response with the density of tumor-infiltrating CD4+ T cells by using the Spearman correlation text (A). The multiplex immunofluorescence images of the tumor sites in patient 17 (B)”

6. The definition of ‘Middle’ and ‘High’ in Figure 1 can be inferred from the text, but is not explicitly given. It would be helpful to explicitly associate them with the CPS values.

Response: Thank you for your kind reminder. We have made some corrections of PD-L1 expression (Middle: $1 \leq \text{PD-L1 combined positive score} \leq 19$; High: $\text{PD-L1 combined positive score} \geq 20$) in Figure 11 below (Figure 2 in the revised manuscript) to make it easier to follow, as shown below.

Figure 11 The waterfall plot of best radiographic response by RECIST 1.1.

7. The text mentions that genetic differences between pCR and Non-pCR are tested and found non-significant, could you add a short description of the testing approach used?

Response: Thank you for your suggestion. Fisher exact test was used for comparisons between pCR and non-pCR groups. We have added the testing approach used in the results section.

“No significant genetic differences were noted between pCR and Non-pCR patients by using the Fisher exact test.” (**Results sections: Page 12 Lines 252-253**)

8. It may be a problem with the rendering, but in Figure 4A pink and blue appear in the plot but not in the legend.

Response: Thank you for your kind reminder. We have updated the Figure 4 to include all the colors in the legend as shown above.

9. It is mentioned that in Figure S4 no distinguishing distribution pattern was identified: also in this case I would report the test, or that this was inferred from visual inspection.

Response: Thank you for your kind reminder. We have provided the test used for analysis in the figure legends, as described above.

10. Figure S5 resolution is quite low.

Response: Thank you for your kind reminder. We have provided all Figures in the PDF

form to ensure the quality.

11. This does not affect the results, but in the adopted Simon's two-stage design the null hypothesis can be rejected if 39 or more patients achieved a CR or PR.

Response: Thank you for your careful review of our manuscript. It should be "the null hypothesis could be rejected if 39 or more patients achieved a CR or PR". We have made the corresponding revision in the statistical analysis of the methods section.

"The null hypothesis could be rejected if 39 or more patients achieved a CR or PR"
(Methods sections: Page 27 Lines 568-569)

REVIEWER COMMENTS

Reviewer #1 (Remarks to the Author):

Thank you for addressing all my queries in the revised manuscript. While I was disappointed in the lack of more tissue to perform mIHC correlative studies, this is still a remarkable study with a pathway to a phase 3 trial, which I hope the team is planning in the future.

Reviewer #2 (Remarks to the Author):

The manuscript is acceptable in the current form.

Reviewer #3 (Remarks to the Author):

Thank you for your detailed answers and for the changes. I only have few residual comments/suggestions:

1. I am still unsure about the variance calculation discussed in point 1 of the revision. The denominator used to obtain the percentages is $n=27$, so $p=17/27$ for MPR or pCR. Then, the corresponding 95% confidence interval bounds, under the normal approximation, should be computed as: $p \pm 1.96 \cdot \sqrt{p \cdot (1-p)/n}$, with $n=27$, consistently with the denominator used for p .

3. Concerning point 3 in the revision, I would not recommend adopting Fisher's exact test in this form, if indeed SD and PD can only be Non-pCR and CR only pCR, as there is no randomness involved for these patients. I would recommend only testing for the non-deterministic associations, i.e., only testing if pCR proportion of PR patients is larger than that of Non-PCR.

5. Concerning the statistical tests:

Figure 4, the assumption of approximate normality may be violated here and I would suggest a Wilcoxon rank sum/Mann-Whitney U nonparametric test instead.

Figure S3: Was this perhaps the Kruskal-Wallis test? In this case I would also rephrase the

text at lines 207-208 as “there was no significant difference in the distributions of TMB or PD-L1 expression levels according to anatomic site or staging”.

Figure S5: A paired test should be used in this situation (Wilcoxon signed-rank test).

The normal approximation for 1-year OS and PFS may be inaccurate given the very small numbers. The log-log confidence interval should be preferred (Borgan, Ø. & Liestøl, K., 1990. A note on confidence intervals and bands for the survival curve based on transformations, *Scandinavian Journal of Statistics* 17, 35–41).

Point-by-point response to the reviewers' comments

Response to Reviewers' comments:

Reviewer #1:

Thank you for addressing all my queries in the revised manuscript. While I was disappointed in the lack of more tissue to perform mIHC correlative studies, this is still a remarkable study with a pathway to a phase 3 trial, which I hope the team is planning in the future.

Response: Thank you for your insightful comments. We are still working on it. Thank you again for your time and kind work on our manuscript.

Reviewer #2:

The manuscript is acceptable in the current form.

Response: Thank you for your time and kind work on our manuscript.

Reviewer #3:

Thank you for your detailed answers and for the changes. I only have few residual comments/suggestions:

1. I am still unsure about the variance calculation discussed in point 1 of the revision. The denominator used to obtain the percentages is $n=27$, so $p=17/27$ for MPR or pCR. Then, the corresponding 95% confidence interval bounds, under the normal approximation, should be computed as: $p \pm 1.96 \cdot \sqrt{p \cdot (1-p)/n}$, with $n=27$, consistently with the denominator used for p .

Response: Thank you for your constructive comments. According to your suggestions, we have recalculated the 95% confidence interval for MPR and pCR and made the corresponding corrections in the abstract and results sections.

"Between April 2021 and March 2022, 48 patients were enrolled and treated. The ORR was 89.6% (95% CI: **80.9**, 98.2) among 48 patients who completed NeoCPC. Of the 27 patients who underwent surgery, 17 (63.0%, 95% CI: **44.7**, **81.2**) achieved a major or complete pathologic response, with a pathologic complete response rate of 55.6% (95% CI: **36.8**, **74.3**)."
(Abstract: Page 3 Lines 45-49)

"Among the 48 patients finally enrolled, 10 achieved a CR (one oral cancer, four oropharyngeal cancer, two laryngeal cancer, and three hypopharyngeal cancer), and 33 achieved a PR (13 oral cancer, eight oropharyngeal cancers,

seven laryngeal cancer, and five hypopharyngeal cancer), with an ORR of 89.6% (95% CI: **80.9, 98.2**) (Figure 2)” (Results: Page 7 Line 144-Page 8 Line 148)

“Of the 27 resected patients, 17 (63.0%, 95% CI: **44.7, 81.2**) achieved an MPR or pCR, with a pCR rate of 55.6% (95% CI: **36.8, 74.3**) (including seven oral cancer, six HPV-positive oropharyngeal cancer, one HPV-negative oropharyngeal cancer, and one laryngeal cancer). Twenty-one out of the 27 patients had a clinical to pathological downstaging (77.8%, 95% CI: **62.1, 93.5**).” (Results: Page 8 Lines 151-155)

3. Concerning point 3 in the revision, I would not recommend adopting Fisher’s exact test in this form, if indeed SD and PD can only be Non-pCR and CR only pCR, as there is no randomness involved for these patients. I would recommend only testing for the non-deterministic associations, i.e., only testing if pCR proportion of PR patients is larger than that of Non-PCR.

Response: Thank you for your constructive comments and suggestions. Among the 18 patients with PR, 11 (61.1%) had pCR and 7 (38.9%) had non-pCR, as shown in **Table 1** below and **Figure 3A**. The pCR proportion of PR patients was numerically larger than that of Non-PCR.

For immunotherapy, sometimes a unique response termed “pseudoprogression” could happen, some patients whose disease met the criteria for disease progression or in stable status based on RECIST were noted to have late but deep and durable responses. So SD and PD based on RECIST has the chance to get pCR in this scenario. However, due to limited sample size, the inconsistency was not observed. We did not perform further association analysis according to your suggestion and sincerely hope it will still meet your approval. Thank you again for your time and kind work on our manuscript.

Table 1 Correlation between radiographic and pathologic response

	CR	PR	ASD	PD
pCR	4	11	0	0
Non-pCR	0	7	3	1

5. Concerning the statistical tests:

(1) Figure 4, the assumption of approximate normality may be violated here and I would suggest a Wilcoxon rank sum/Mann-Whitney U nonparametric test instead.

Response: Thank you for your constructive comments. We have used Mann-Whitney U nonparametric test instead according to your suggestions. We have updated the P values and corresponding markers in the main text and Figures 4C and 4G, respectively, as well as the figure legend.

“Significantly, a higher radiographical response rate was observed in HPV-positive patients ($p=0.012$, Figure 4B). Meanwhile, a lower radiographical response rate was observed in patients with altered TP53 ($p=0.006$, Figure 4C) and those with altered TERT ($p=0.01$, Figure 4D). The correlation analyses of TP53/TERT alternations with HPV were further performed. As shown in Figure 4E, the TP53 alternation was more common in HPV-positive patients ($p<0.001$), while the TERT alternation was also more common in HPV-positive patients, although the difference did not reach statistical significance ($p=0.16$). The correlations of radiographical tumor response with TP53/TERT alternations were further carried out among the HPV-positive patients. **TP53 alternation was not significantly associated with radiographic tumor response ($p=0.807$, Figure 4F). A lower radiographic response rate was observed in patients with altered TERT ($p=0.049$, Figure 4G)**” (Results: Page 11 Lines 213-223)

Figure 4 The genetic landscape and radiographic tumor response analysis. Alterations as assessed by next-generation sequencing of baseline primary tumor samples (A); Comparisons of radiographic tumor response between HPV-positive and HPV-negative patients (B); TP53-mutant and TP53-wildtype patients (C); and TERT-mutant and TERT-wildtype patients by using the **Mann-Whitney U nonparametric test** (D). Correlation between HPV status and TP53 or TERT mutations by using the **Mann-Whitney U nonparametric test** (E). Comparisons of radiographic tumor response

between TP53-mutant and TP53-wildtype patients (F); and TERT-mutant and TERT-wildtype patients (G) in HPV-negative patients by using the **Mann-Whitney U nonparametric test. The histogram plots show the mean values of radiographic tumor response and standard deviation (SD). The dot represents an individual data point.** *: $p < 0.05$, **: $p < 0.01$, ***: $p < 0.001$, ns: No significance. Source data are provided as a Source Data file." (**Figure legends**)

(2) Figure S3: Was this perhaps the Kruskal-Wallis test? In this case I would also rephrase the text at lines 207-208 as "there was no significant difference in the distributions of TMB or PD-L1 expression levels according to anatomic site or staging".

Response: Thank you for your comments. We have used Kruskal-Wallis test according to the suggestion. The distribution of PD-L1 expression was significantly associated with the different sites of tumor origins ($p = 0.036$). There was no significant difference in the distributions of TMB or PD-L1 expression levels according to TNM staging or in the distribution of PD-L1 expression according to anatomic site. We have rephrased the sentence and made corresponding revisions in the Figure and legend.

"The distribution of PD-L1 expression was significantly associated with the different sites of tumor origins ($p = 0.036$). There was no significant difference in the distributions of TMB or PD-L1 expression levels according to TNM staging or in the distribution of PD-L1 expression according to anatomic site (Figure S3)" (Results: Page 10 206-209)

"Figure S3 Distribution of TMB and PD-L1 expression level in patients with different anatomic sites and TNM stages analyzed by using the **Kruskal-Wallis test. The dot represents an individual data point. The center line with a circle represents mean value, and the bar represents standard deviation.** *: $p < 0.05$, ns: No significance. Source data are provided as a Source Data file." (**Figure legends**)

(3) Figure S5: A paired test should be used in this situation (Wilcoxon signed-rank test).

Response: Thank you for your constructive comments. We have used the paired Wilcoxon signed-rank test according to your suggestions. M1-like macrophage cells in the tumor stroma were significantly increased after treatment in overall patients ($p = 0.0289$), primarily in patients with pCR ($p = 0.0234$), but not in those with non-pCR ($p > 0.05$). We have made corresponding revisions in the main text and figure legends.

"M1-like macrophage cells in the tumor stroma were significantly increased

after treatment in patients with pCR (**p=0.0234**), but not in those with non-pCR (**p>0.05**). The other types of tumor infiltration immune cells were not significantly changed (Figure S5).” (**Results: Page 12 Lines 249-251**)

“**Figure S5** Comparisons of the density of tumor infiltration immune cells before and after neoadjuvant therapy in overall patients (A) and those with pCR (B) and non-pCR (C) using the **paired Wilcoxon signed-rank test**. **The blue dot represents an individual data point of cell density at baseline, the red dot represents an individual data point after neoadjuvant therapy. The black line represents the direction of change in cell density following neoadjuvant therapy.** *: p<0.05, ns: No significance. **Source data are provided as a Source Data file.**” (**Figure legends**)

(4) The normal approximation for 1-year OS and PFS may be inaccurate given the very small numbers. The log-log confidence interval should be preferred (Borgan, Ø. & Liestøl, K., 1990. A note on confidence intervals and bands for the survival curve based on transformations, Scandinavian Journal of Statistics 17, 35–41).

Response: Thank you for your constructive comments. We have used the Log-Log transformation method to estimate the 95% confidence interval. The estimated one-year OS and PFS rates were both 97.9% (95% CI: 86.1, 99.7). We have made the corresponding revisions in the results and methods sections.

“The estimated one-year OS and PFS rates were both 97.9% (95% CI: **86.1, 99.7**).” (**Results: Page 8 Lines 165-166**)

“The PFS and OS were estimated using the Kaplan-Meier method **and the corresponding 95% confidence intervals (CIs) were estimated with the Log-Log transformation method.**” (**Methods: Page 26 Lines 552-554**)

REVIEWERS' COMMENTS

Reviewer #3 (Remarks to the Author):

Thank you for your detailed answers.

About point 3, I would then suggest to only remove the CR cases from the hypothesis testing - if again my understanding is correct that they can only be pCR. Also, an appropriate test for agreement is Cohen's kappa.

Point-by-point response to the reviewers' comments

Response to Reviewers' comments:

Reviewer #3:

Thank you for your detailed answers.

About point 3, I would then suggest to only remove the CR cases from the hypothesis testing - if again my understanding is correct that they can only be pCR. Also, an appropriate test for agreement is Cohen's kappa.

Response: Thank you for your further comments. Theoretically, a radiographic CR may not always be a pCR. The radiographic CR is defined as the disappearance of all target lesions, and any pathological lymph nodes, whether target or non-target, must have reduction in a short axis to <10 mm, according to the RECIST version 1.1 criteria ^[1]. That is, lymph node regression to <10 mm can be defined as CR. For example, the normal tonsil can be identified on CT, and when the tonsillar mass on the affected side regresses to the size of a normal tonsil, we can also rate it as a CR. Therefore, it is difficult to say that CR is always pCR, after all, the pathology is more subtle, while imaging itself may have some limitations. In our study, all patients with a CR achieved a pCR, while our sample size is small.

Accordingly, we performed the Cohen's kappa test in the overall patients and those without a CR. The kappa value was 0.397 and 0.364, respectively, as shown in Table 1 and 2 below. However, our study was underpowered to find a significant correlation between radiographic and pathologic response in either overall population or those without a CR.

Table 1 Correlation between radiographic and pathologic response in the overall patients

	Radiographic response		Kappa
	ORR	Non-ORR	
Pathological response, n			0.397

pCR	15	0
Non-pCR	7	4

Table 1 Correlation between radiographic and pathologic response in patients without a CR

	Radiographic response		Kappa
	PR	SD/PD	
Pathologic response, n			0.364
pCR	11	0	
Non-pCR	7	4	

[1] Eisenhauer EA, Therasse P, Bogaerts J, et al. New response evaluation criteria in solid tumours: revised RECIST guideline (version 1.1). Eur J Cancer. 2009;45(2):228-247. doi:10.1016/j.ejca.2008.10.026.